# MMFakeBench: A Mixed-Source Multimodal Misinformation Detection Benchmark for LVLMs

**Xuannan Liu**[1]    **Zekun Li**[2]    **Peipei Li**[1*]    **Huaibo Huang**[3]    **Shuhan Xia**[1]
**Xing Cui**[1]    **Linzhi Huang**[1]    **Weihong Deng**[1]    **Zhaofeng He**[1]
[1]Beijing University of Posts and Telecommunications    [2]University of California, Santa Barbara
[3]Center for Research on Intelligent Perception and Computing, NLPR, CASIA
{liuxuannan, lipeipei}@bupt.edu.cn    zekunli@cs.ucsb.edu
https://liuxuannan.github.io/MMFakeBench.github.io/

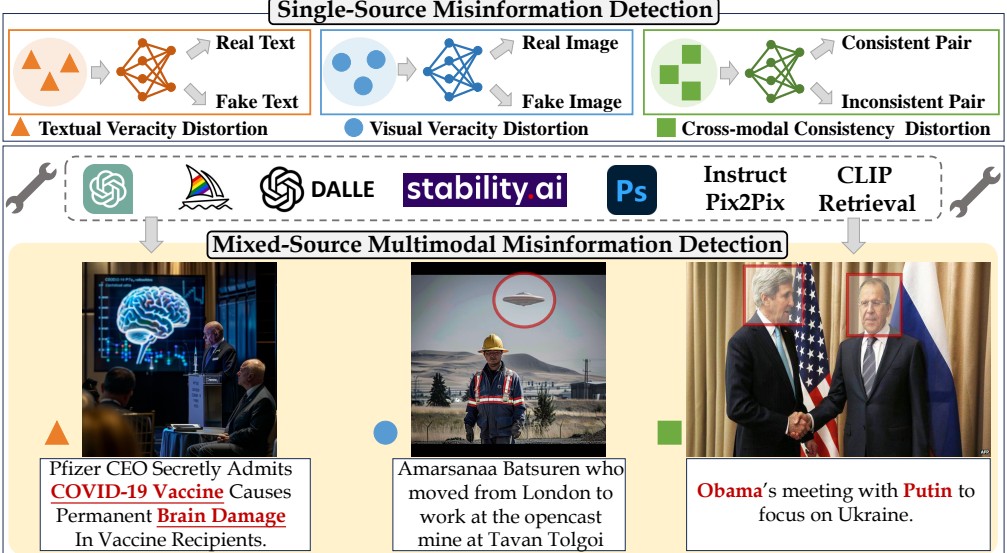

Figure 1: **Top:** Previous methods often assume a single misinformation source and conduct single-source detection. **Bottom:** We collaborate generative models and AI tools to build a mixed-source multimodal misinformation benchmark and achieve mixed-source detection.

## Abstract

Current multimodal misinformation detection (MMD) methods often assume a single source and type of forgery for each sample, which is insufficient for real-world scenarios where multiple forgery sources coexist. The lack of a benchmark for mixed-source misinformation has hindered progress in this field. To address this, we introduce MMFakeBench, the first comprehensive benchmark for mixed-source MMD. MMFakeBench includes 3 critical sources: textual veracity distortion, visual veracity distortion, and cross-modal consistency distortion, along with 12 sub-categories of misinformation forgery types. We further conduct an extensive evaluation of 6 prevalent detection methods and 15 Large Vision-Language Models (LVLMs) on MMFakeBench under a zero-shot setting. The results indicate that current methods struggle under this challenging and realistic mixed-source MMD setting. Additionally, we propose MMD-Agent, a novel approach to integrate the reasoning, action, and tool-use capabilities of LVLM agents, significantly enhancing accuracy and generalization. We believe this study will catalyze future research into more realistic mixed-source multimodal misinformation and provide a fair evaluation of misinformation detection methods.

---

*Corresponding Author

# 1 INTRODUCTION

Recent advances in generative models for texts (Brown et al., 2020; Touvron et al., 2023) and images (Dhariwal & Nichol, 2021; Cui et al., 2024b;a) have significantly lowered the barrier to producing diverse multimodal misinformation, posing threats to politics, finance, and public health. For instance, the misinformation "COVID-19 vaccine causes brain damage", shown in Fig. 1, accompanied by a highly convincing image, can lead to public distrust in medical treatments and vaccine refusal. Therefore, identifying multimodal misinformation on social media is urgent.

Most current multimodal misinformation detection (MMD) methods (Abdelnabi et al., 2022; Qi et al., 2024; Ying et al., 2023; Huang et al., 2023; Zhang & Gao, 2023; Lee et al., 2021) typically assume that each sample has a single, known forgery source. As depicted in Fig. 1 **Top**, these forgery sources involve either textual veracity with fake news text, visual veracity with fake images, or inconsistency between the text and image. However, the single-source assumption is overly simplistic and fails to capture the complexity of real-world scenarios, where misinformation often stems from multiple, random sources. To address this mixed-source MMD problem, two key challenges need to be solved. First, existing datasets primarily consist of single-source misinformation, lacking misinformation from multiple sources. This limitation prevents comprehensive evaluation of MMD methods. Second, there is a lack of general detectors capable of handling mixed-source misinformation. Hence, we present MMFakeBench, encompassing the mixed-source MMD benchmark, evaluations, and framework.

**Benchmark:** We introduce MMFakeBench, the first comprehensive benchmark for evaluating mixed-source MMD. As shown in Fig. 1 **Bottom**, leveraging advanced AI tools, such as diffusion generators and ChatGPT, MMFakeBench provides 12 forgery types with 11,000 data pairs from three primary sources: ***textual veracity distortion***, ***visual veracity distortion***, and ***cross-modal consistency distortion***. Textual veracity distortion encompasses three types of rumors: natural, artificial, and GPT-generated rumors. Unlike (Thorne et al., 2018; Shu et al., 2020; Hanselowski et al., 2019; Chen & Shu, 2024) that focus solely on single-source, text-only rumors, MMFakeBench incorporates text-image rumors using highly relevant real or AI-generated images. Visual veracity distortion filters existing PS-edited images (Da et al., 2021; Nakamura et al., 2020) according to misinformation standards and incorporates high-quality AI-generated images. Cross-modal consistency distortion integrates inconsistencies from both edited and repurposed perspectives into five distinct sub-categories.

**Evaluations:** To access the current advancements in mixed-source MMD, we build the fine-grained multi-class evaluation metric and conduct a comprehensive evaluation of 6 state-of-the-art detection methods and 15 large vision-language models (LVLMs) on MMFakeBench. Specifically, we evaluate 6 detection methods in a single-source setting and assess their combined performance (text, image, and cross-modal inconsistency detectors) in the mixed-source setting. Additionally, we evaluate 15 large vision-language models (LVLMs), including proprietary models such as GPT-4V (OpenAI, 2023). The results indicate that existing detection methods exhibit poor generalization. Although LVLMs show robust generalization capabilities, their overall performance still requires improvement.

**Framework:** Based on our analysis, we propose a simple yet effective LVLM-based framework called **MMD-Agent**, which enhances detection performance and serves as a new baseline for future research. MMD-Agent decomposes mixed-source detection into three stages: textual veracity check, visual veracity check, and cross-modal consistency reason. This decomposition ensures methodical and thorough reasoning. At each stage, MMD-Agent instructs LVLMs to generate multi-perspective reasoning traces, integrating model actions for coherent decisions. Additionally, the models interact with external knowledge sources via tools (e.g., Wikipedia) to incorporate supplementary information into their reasoning.

In summary, the main contributions are: (1) We introduce mixed-source multimodal misinformation detection (MMD), a challenging setting for detecting misinformation from diverse and uncertain sources, breaking free from single-source constraints, and advancing practical misinformation detection tasks. (2) We develop MMFakeBench, the first benchmark dataset for evaluating mixed-source MMD. The dataset contains 3 critical categories (textual veracity distortion, visual veracity distortion, and consistency reasoning) and 12 sub-categories of forgery types. (3) Using the newly collected dataset, we benchmark mixed-source MMD by evaluating 6 prevalent detection methods and 15 LVLMs. (4) We propose MMD-Agent, a simple yet effective LVLM-based framework. It outperforms previous methods and LVLMs on the MMFakeBench benchmark, highlighting the potential of mixed-source MMD and providing a new baseline for future research.

Table 1: Comparison of misinformation datasets. (k) denotes the number of rumor types.

| Dataset | Textual Veracity Distortion | | | Visual Veracity Distortion | | | Cross-modal Consistency Distortion | |
|---|---|---|---|---|---|---|---|---|
| | Text | Supporting Image | | Text | Fact-conflicting Image | | Image/Text | Image/Text |
| | (Rumor) | Repurposed | AI-generated | (Veracity) | PS-edited | AI-generated | Repurposing | Editing |
| FEVER (Thorne et al., 2018) | ✔ (1) | ✗ | ✗ | ✗ | ✗ | ✗ | ✗ | ✗ |
| Politifact (Shu et al., 2020) | ✔ (1) | ✗ | ✗ | ✗ | ✗ | ✗ | ✗ | ✗ |
| Gossipcop (Shu et al., 2020) | ✔ (1) | ✗ | ✗ | ✗ | ✗ | ✗ | ✗ | ✗ |
| Snopes (Hanselowski et al., 2019) | ✔ (1) | ✗ | ✗ | ✗ | ✗ | ✗ | ✗ | ✗ |
| MOCHEG (Yao et al., 2023) | ✔ (1) | ✗ | ✗ | ✗ | ✗ | ✗ | ✗ | ✗ |
| LLMFake (Chen & Shu, 2024) | ✔ (1) | ✗ | ✗ | ✗ | ✗ | ✗ | ✗ | ✗ |
| EMU (Da et al., 2021) | ✗ | ✗ | ✗ | ✗ | ✔ | ✗ | ✗ | ✗ |
| Fakeddit (Nakamura et al., 2020) | ✗ | ✗ | ✗ | ✔ | ✔ | ✗ | ✗ | ✗ |
| MAIM (Jaiswal et al., 2017) | ✗ | ✗ | ✗ | ✗ | ✗ | ✗ | ✔ | ✗ |
| MEIR (Sabir et al., 2018) | ✗ | ✗ | ✗ | ✗ | ✗ | ✗ | ✗ | ✔ |
| NewsCLIPpings (Luo et al., 2021) | ✗ | ✗ | ✗ | ✗ | ✗ | ✗ | ✔ | ✗ |
| COSMOS (Aneja et al., 2023) | ✗ | ✗ | ✗ | ✗ | ✗ | ✗ | ✔ | ✗ |
| DGM4 (Shao et al., 2023) | ✗ | ✗ | ✗ | ✗ | ✗ | ✗ | ✗ | ✔ |
| **MMFakeBench (Ours)** | ✔ (3) | ✔ | ✔ | ✔ | ✔ | ✔ | ✔ | ✔ |

## 2 RELATED WORK

**Misinformation Benchmarks.** One group of misinformation datasets primarily focuses on distorting textual veracity. The FEVER (Thorne et al., 2018) dataset is constructed by manipulated Wikipedia sentences with manual annotation. Unlike these artificial rumors, other datasets, such as Snopes (Hanselowski et al., 2019), Politifact, Gossipcop (Shu et al., 2020), and MOCHEG (Yao et al., 2023), collect natural rumors from fact-checking websites. Recently, the LLMFake (Chen & Shu, 2024) instructs large language models (LLMs) to generate diverse misinformation. Apart from misleading text, the EMU (Da et al., 2021) and Fakeddit (Nakamura et al., 2020) collect Photoshop-edited images from the Reddit platform. Another group of misinformation datasets focuses on disrupting cross-modal consistency. The MAIM (Jaiswal et al., 2017) and MEIR (Sabir et al., 2018) datasets employ caption replacement and entity swapping, respectively. The NewsCLIPpings (Luo et al., 2021) and COSMOS (Aneja et al., 2023) datasets link out-of-context images to support certain narratives. The recent dataset DGM[4] (Shao et al., 2023) introduces global and local manipulation to alter semantics and sentiment. Different from these works containing only single-source misinformation, we propose the first benchmark dataset for evaluating mixed-source MMD, involving textual veracity distortion, visual veracity distortion, and cross-modal consistency distortion, shown in Table 1.

**Misinformation Detection.** Current misinformation detection approaches are mainly divided into two categories. The first is to check textural veracity by constructing features based on writing style (Przybyla, 2020), sentiment (Ghanem et al., 2021), user feedback (Min et al., 2022) and pre-trained language models (Huang et al., 2023; Zhang & Gao, 2023). The second is to fuse cross-modal features to detect semantic inconsistencies. Previous works focus on devising attention-based modules (Qian et al., 2021b; Ying et al., 2023; Wu et al., 2021) guided by diverse learning strategies (Chen et al., 2023). Recent works Shao et al. (2024); Qi et al. (2024); Liu et al. (2024c) capitalize the VLMs which benefit from large-scale pre-training for reasoning context cues. However, these works target a single-source problem, and evaluations are conducted in constrained scenarios. Our work is the first to introduce a comprehensive benchmark for mixed-source multimodal misinformation detection.

**Large Vision-Language Models.** Large language models (LLMs) such as GPT-3 (Brown et al., 2020) and Vicuna (Chiang et al., 2023) have demonstrated remarkable performance on various linguistic tasks. Inspired by LLMs, models like LLaVA (Liu et al., 2023a) and MiniGPT-4 (Zhu et al., 2023) facilitate image-text feature alignment by leveraging visual instruction tuning. More recently, the evolution of LVLMs has driven advancements in creating diverse and high-quality multimodal instruction datasets. Models such as InstructBLIP (Dai et al., 2023), mPLUG-Owl (Ye et al., 2023; 2024), LLaVA-1.5 (Liu et al., 2024a) exemplify these developments. In this paper, we explore the reasoning capabilities (Zheng et al., 2023; Zhang et al., 2024) of LVLMs to address the challenge of mixed-source multimodal misinformation by integrating reasoning, actions, and tool-use capabilities.

## 3 MMFAKEBENCH BENCHMARK

In MMFakeBench, we focus on multimodal misinformation involving both text and images, categorizing it into three distinct types based on the sources of falsified content:

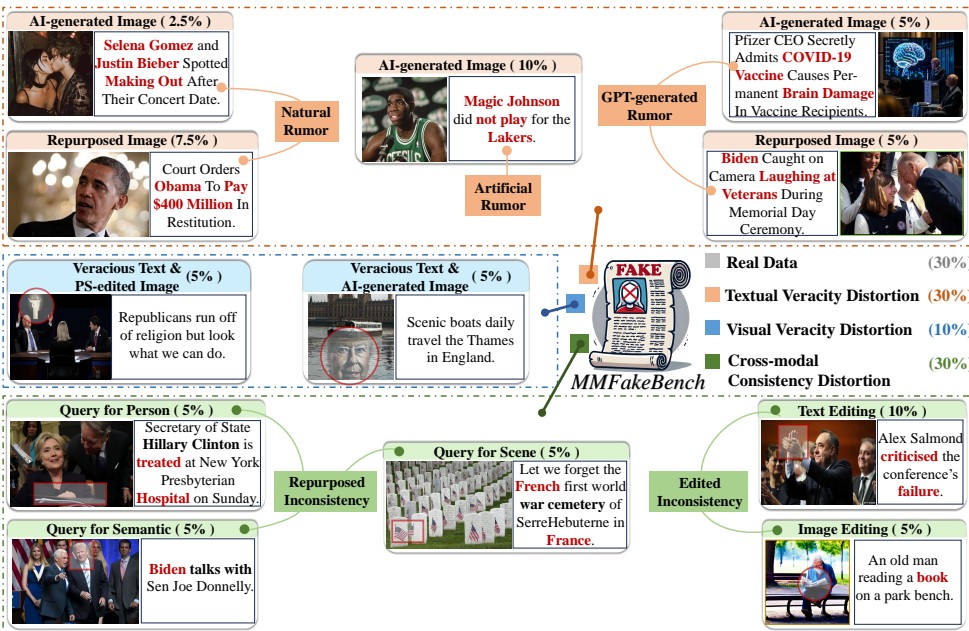

Figure 2: Statistics of the MMFakeBench Benchmark.

• *Textual Veracity Distortion.* It incorporates text-based rumors paired with supporting images to mimic real-world multimodal scenarios. An image that visually supports misleading text can make the misinformation appear more credible and persuasive to the users.

• *Visual Veracity Distortion.* Images in this category contain fact-conflicting misinformation through altered or fabricated elements, while the texts remain veracious. These visual manipulations often lead people to perceive falsified content as authentic, distorting their understanding of the information.

• *Cross-modal Consistency Distortion.* Even when the text and image are individually correct, their combination can generate potential misinterpretation if presented in a manner that introduces incorrect associations or semantic discrepancies between the two modalities, thereby misleading people.

## 3.1 THREE MISINFORMATION SOURCES

### 3.1.1 TEXTUAL VERACITY DISTORTION

Textual veracity distortion is a critical misinformation category. Our dataset in this category comprises 3,300 samples. Previous works (Thorne et al., 2018; Shu et al., 2020; Hanselowski et al., 2019; Chen & Shu, 2024) focus on single-source and single-modal textual rumors. However, the types of real-world rumors are diverse, and those accompanied by images can have a significantly greater impact. To address this, MMFakeBench introduces a broader range of rumor types and augments them with highly relevant supporting images to enhance perceived credibility.

**Textual Rumor.** As shown in Table 1, unlike previous methods that consider only one type of textual rumor, we consider three types: (1) Natural Rumor. We select natural rumors from Politifact and Gossipcop (Shu et al., 2020), which provide political news and entertainment stories derived from fact-checking websites. (2) Artificial Rumor. We collect artificial rumors from the FEVER dataset (Thorne et al., 2018), which is curated by manually modifying Wikipedia sentences. (3) GPT-generated Rumor. We instruct ChatGPT (`gpt-3.5-turbo`) to produce rumors via three prompt approaches (Chen & Shu, 2024): arbitrary generation, rewriting generation, and information manipulation. Arbitrary generation is utilized to generate misinformation in specific domains. Rewriting generation addresses the concise and synthetic traces of artificial rumors, while information manipulation involves altering factual information in real claims from Politifact and Gossipcop.

**Supporting Image.** We use either AI-generated images or carefully selected real images to support the content presented in the rumor text. (1) AI-generated Image: For artificial rumors and their derived GPT-generated rumors, as well as some less harmful gossip, we utilize generative models to

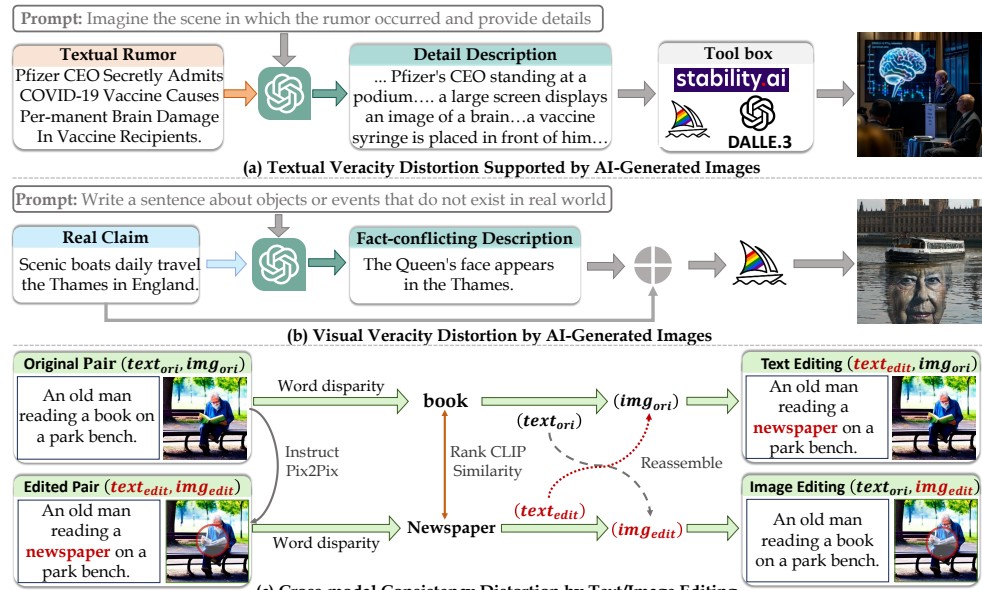

Figure 3: Illustrations of using collaborative generative models and AI tools to generate different sources of misinformation.

create supporting images. We utilize three advanced models: Stable Diffusion XL (Rombach et al., 2022), DALL-E3 (Ramesh et al., 2022), and Midjourney V6 (Midjourney, 2022) to enhance the diversity of the generated images. For each rumor, we randomly select a generative model to produce a corresponding supporting image. As many rumors are highly abstract and lack concrete descriptions of objects and scenes, directly using these texts as conditions often yields images that are neither realistic nor relevant. To address this, as shown in Fig. 3 (a), we instruct ChatGPT to enrich the rumors with more detailed descriptions. These enriched contexts are then used as input for generative models, ensuring alignment with the textual rumors. (2) Repurposed Image: To avoid creating new high-risk images, especially for sensitive topics like politics and gossip, we use repurposed images from the VisualNews (Liu et al., 2021) dataset, which contains numerous image-text pairs from real-world news sources. We select images with high semantic relevance to the textual rumors based on text-image CLIP similarity and text-text CLIP similarity. Images with the highest similarity scores are chosen as supporting images.

### 3.1.2 VISUAL VERACITY DISTORTION

The visual veracity distortion dataset comprises 1,100 samples where the text is real and the misinformation exists in the image. Previous datasets (Da et al., 2021; Nakamura et al., 2020) focus solely on PS-edited (Photoshop-edited) images, containing both misleading and non-misleading content. In this study, we manually select the misleading ones and include them in MMFakeBench. Besides, we incorporate AI-generated images with veracity distortion, which is increasingly harmful due to advancements in diffusion generators.

**PS-Edited Image.** The PS-edited images are derived from the "manipulated content" samples in the Fakeddit dataset (Nakamura et al., 2020), which is designed for multimodal fake news detection. These samples originate from the "Photoshop battles comments" on Reddit. PS-edited samples in the Fakeddit typically exhibit either aesthetic modifications or fact-conflicting manipulations. Since aesthetic modifications do not compromise the factuality of the visual content, they are excluded from misinformation standards. Consequently, ten of the volunteers participate in selecting 550 PS-edited images containing fact-conflicting content from the 7,693 samples of the "manipulated content" set.

**AI-generated Image.** We propose an automated pipeline that generates fact-conflicting descriptions from text captions and then creates high-quality images. Specifically, we first collect image-text pairs from the MS-COCO (Lin et al., 2014) and VisualNews datasets (Liu et al., 2021). Based on the original text captions, we use ChatGPT to generate corresponding fact-conflicting descriptions, which are manually verified, as depicted in Fig. 3 (b). For example, from the caption "Scenic boats daily travel the Thames in England", we generate the description "The Queen's face appears in

the Thames". These descriptions, combined with the original captions, are used as prompts in the Midjourney V6 model (Midjourney, 2022) to create corresponding images. The resulting text-image pairs contain original factual text and generated images with additional fact-conflicting information.

### 3.1.3 CROSS-MODAL CONSISTENCY DISTORTION

In cross-modal consistency distortion, both the text and image with veracity, but either the text or image is replaced/manipulated to disrupt their overall consistency. Previous datasets (Sabir et al., 2018; Luo et al., 2021; Shao et al., 2023) focus on inconsistencies from a single source, either edit-based or repurposed-based. In contrast, our MMFakeBench integrates inconsistencies from both edited and repurposed perspectives into five distinct sub-categories, a total of 3,300 image-text pairs.

**Repurposed Inconsistency.** Our dataset contains three types of repurposed inconsistency, curated directly from the NewsCLIPings (Luo et al., 2021) dataset: semantic query, person query, and scene query. (1) Semantic query retrieves repurposed images based on specific semantic content. (2) Person query ensures the individual mentioned in the caption appears in the mismatched image. (3) Scene Query relies on spatial similarity to retrieve comparable scene information from repurposed images.

**Edited Inconsistency.** Our dataset contains two types of edited inconsistency: text editing and image editing. For text editing, we select samples from the DGM[4] (Shao et al., 2023) dataset, which modifies sentiment words with their antonyms. Notably, some samples in DGM[4] contain fact-conflicting content post-editing. To avoid redundancy with textual veracity distortion, we filter out these samples. For remaining text editing and all image editing inconsistencies, we build upon the COCO-Counterfactuals (Le et al., 2023) dataset. This dataset encompasses original image-text pairs $(text_{ori}, img_{ori})$ and edited image-text pairs $(text_{edit}, img_{edit})$ which are obtained via Instruct-Pix2Pix model (Brooks et al., 2023). As illustrated in Fig. 3 (c), we separately extract word disparities between $text_{ori}$ and $text_{edit}$ and select samples with significant semantic differences using CLIP similarity. Then, we reassemble the two pairs and obtain $(text_{edit}, img_{ori})$ as text-edited consistency distortion samples and $(text_{ori}, img_{edit})$ as image-edited consistency distortion samples.

### 3.2 REAL DATA COLLECTION

In addition to the misinformation data, we collect 3,300 real data pairs, ensuring both textual and visual veracity and exhibiting strong image-text consistency. Given that our synthetic data is derived from multiple datasets, we construct the real dataset from the same corresponding sources, including MS-COCO, VisualNews, and real image-text pairs from Fakeddit. We further divide VisualNews into four distinct news sources: The Guardian, BBC, USA TODAY, and The Washington Post. Finally, we build the real dataset by equally selecting from six distinct sources.

### 3.3 MMFAKEBENCH ANALYSIS

MMFakeBench consists of 11,000 image-text pairs, which are divided into a validation set and a test set following (Yue et al., 2024). The validation set, comprising 1,000 image-text pairs, is intended for hyperparameter selection, while the test set contains 10,000 pairs. MMFakeBench encompasses one real category and three misinformation categories. Detailed statistics are shown in Fig. 2. MMFakeBench is partitioned into 30% for textual veracity distortion, 10% for visual veracity distortion, 30% for cross-modal consistency distortion, and 30% for real data. The three misinformation categories can be further subdivided into 12 detailed subcategories based on the sources of the text and images in Fig. 2. Such a comprehensive benchmark highlights the challenges of intertwining mixed-source and multiple-types multimodal misinformation in the real world.

## 4 MMD-AGENT FRAMEWORK

We present a simple yet effective framework, *MMD-Agent*, which integrates the reasoning, actions, and tool-use capabilities of LVLM agents. As shown in Fig. 4, MMD-Agent involves two main processes: (1) Hierarchical decomposition and (2) Integration of internal and external knowledge.

Specifically, we instruct LVLMs $\mathcal{M}$ to decompose the task of mixed-source multimodal misinformation detection into three smaller subtasks: textual veracity check, visual veracity check, and cross-modal consistency reasoning. During the intermediary phase, each subtask $t$ is addressed

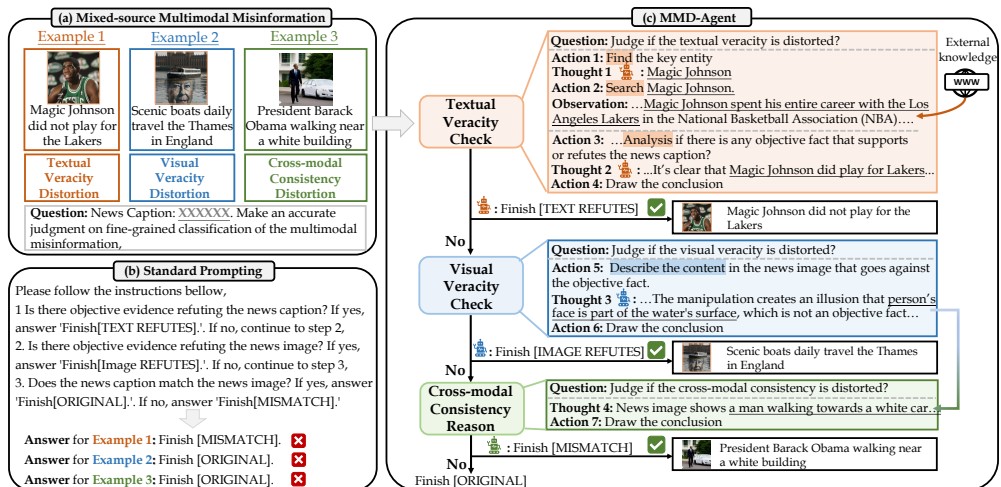

Figure 4: Comparison of standard prompting and proposed MMD-Agent. (a) Three examples of multimodal misinformation from distinct sources. (b) LVLMs with standard prompting methods fail to make correct judgments. (c) MMD-Agent instructs LVLMs to decompose mixed-source detection into smaller subtasks, which are solved by integrating model thoughts and environment observation.

through an interleaved sequence of reasoning and action. The LVLM is guided to reason and induce the needed action to solve the task. The actions $a_t$ are then executed either by leveraging the model's internal knowledge to generate "Thought" $\mathcal{R}_t$ or by interacting with external sources to gather additional information ("Observation") $\mathcal{O}_t$. These action outputs will be integrated into the sequence to facilitate subsequent decision-making $d_t$:

$$d_t = \mathcal{M}\left(\mathcal{R}_t, \mathcal{O}_t, a_t\right). \tag{1}$$

The LVLM's internal capabilities and knowledge are utilized both to reason about which actions to take and to perform those actions from various perspectives, such as identifying key textual entities (Thought 1), conducting factual analysis (Thought 2), and applying commonsense reasoning (Thought 3). However, the model may experience hallucinations when relying solely on its internal knowledge. To address this, we enable the model to interact with external knowledge bases, such as the Wikipedia API, to retrieve reliable and up-to-date information for fact-checking. This approach ensures the accuracy and relevance of the knowledge used in verification.

## 5 EXPERIMENTS

We select 6 state-of-the-art misinformation detection methods and 15 large vision-language models (LVLMs) for preliminary benchmarking using the MMFakeBench dataset, aiming to explore their applicability in the mixed-source MMD setting. Additionally, we evaluate the performance of our proposed framework, MMD-Agent. All evaluations are conducted under a zero-shot setting on our benchmark. All experiments are performed on eight NVIDIA GeForce 3090 GPUs with PyTorch.

### 5.1 EXPERIMENTAL SETTING

**Baseline Models.** We select LVLMs of varying sizes as baseline models. (i) LVLMs with 7B parameter including Otter (Li et al., 2023a), MiniGPT-4 (Zhu et al., 2023), InstructBLIP (Dai et al., 2023), Qwen-VL (Bai et al., 2023), VILA (Lin et al., 2024), PandaGPT (Su et al., 2023), mPLUG-Owl2 (Ye et al., 2024), BLIP-2 (Li et al., 2023b) and LLaVA-1.6 (Liu et al., 2024b). (ii) LVLMs with 13B parameter including VILA, InstructBLIP, BLIP-2, and LLaVA-1.6. (iii) LVLMs with 34B parameter including LLaVA-1.6. (iv) Closed-source model including GPT-4V (OpenAI, 2023).

**Evaluation Metrics.** We evaluate the performance of different baselines using multi-class classification, categorizing data into four distinct classes: textual veracity distortion, visual veracity distortion, cross-modal consistency distortion, and real class. Consistent with (Qian et al., 2021a; Zhang & Gao, 2023; Chen et al., 2023), we adopt the widely-used macro-F1 metric, which balances precision and recall through a harmonic mean. Beyond the F1 score, we also include macro-precision, macro-recall,

Table 2: Overall results (%) of different models on the MMFakeBench validation and test set with the comparison of standard prompting (Standard) and proposed MMD-Agent framework.

| Model | Language | Prompt | Validation (1000) | | | | Test (10000) | | | |
| Name | Model | Method | F1↑ | Precision↑ | Recall↑ | ACC↑ | F1↑ | Precision↑ | Recall↑ | ACC↑ |
|---|---|---|---|---|---|---|---|---|---|---|
| *Human Evaluation* | | | 35.9 | 38.3 | 38.9 | 37.9 | - | - | - | - |
| **LVLMs with 7B Parameter** | | | | | | | | | | |
| Otter-Image | MPT-7B | Standard | 5.2 | 10.5 | 3.4 | 4.1 | 4.9 | 9.3 | 3.3 | 4.0 |
| MiniGPT4 | Vicuna-7B | Standard | 5.2 | 5.2 | 21.2 | 9.0 | 5.3 | 6.9 | 21.0 | 9.1 |
| InstructBLIP | Vicuna-7B | Standard | 7.1 | 7.9 | 6.5 | 7.8 | 8.1 | 16.4 | 7.2 | 8.5 |
| Qwen-VL | Qwen-7B | Standard | 7.5 | 10.3 | 24.3 | 11.0 | 8.0 | 35.9 | 25.5 | 11.6 |
| VILA | LLaMA2-7B | Standard | 11.5 | 7.5 | 25.0 | 30.0 | 11.5 | 7.5 | 25.0 | 30.0 |
| PandaGPT | Vicuna-7B | Standard | 11.8 | 9.8 | 25.0 | 30.0 | 11.6 | 8.6 | 25.0 | 30.0 |
| mPLUG-Owl2 | LLaMA2-7B | Standard | 14.5 | 22.2 | 25.9 | 31.1 | 15.1 | 25.2 | 26.3 | 31.5 |
| BLIP2 | FlanT5-XL | Standard | 16.4 | 20.1 | 27.5 | 33.0 | 16.7 | 17.3 | 27.7 | 33.2 |
| LLaVA-1.6 | Vicuna-7B | Standard | 17.4 | 14.8 | 25.7 | 30.8 | 19.0 | 16.5 | 26.9 | 32.3 |
| **LVLMs with 13B Parameter** | | | | | | | | | | |
| VILA | LLaMA2-13B | Standard | 11.5 | 7.5 | **25.0** | 30.0 | 11.6 | **32.5** | 25.0 | **30.0** |
| | | MMD-Agent | **22.7** | **27.3** | 24.4 | 28.7 | **24.0** | 30.4 | **25.5** | 29.4 |
| InstructBLIP | Vicuna-13B | Standard | 13.7 | 13.2 | 24.0 | 28.8 | 13.9 | 25.5 | 24.3 | **29.1** |
| | | MMD-Agent | **26.0** | **33.3** | **30.1** | **29.5** | **24.5** | **32.1** | **28.8** | 27.3 |
| BLIP2 | FlanT5-XXL | Standard | 16.7 | 34.9 | 27.3 | 32.8 | 16.3 | 34.6 | 27.3 | **32.8** |
| | | MMD-Agent | **31.6** | **39.8** | **32.2** | **34.4** | **28.8** | **39.0** | **30.4** | 32.1 |
| LLaVA-1.6 | Vicuna-13B | Standard | 12.0 | 22.5 | 25.0 | 30.0 | 14.4 | 35.7 | 26.0 | 31.2 |
| | | MMD-Agent | **38.0** | **44.5** | **41.0** | **40.6** | **34.5** | **42.7** | **37.5** | **37.4** |
| **LVLMs with 34B Parameter** | | | | | | | | | | |
| LLaVA-1.6 | Nous-Hermes-2 | Standard | 25.7 | 44.5 | 33.7 | 40.4 | 25.4 | 44.1 | 33.8 | 40.5 |
| | -Yi-34B | MMD-Agent | **49.9** | **54.4** | **52.9** | **48.7** | **47.7** | **52.1** | **49.6** | **46.6** |
| **Proprietary LVLMs** | | | | | | | | | | |
| GPT-4V | ChatGPT | Standard | 51.0 | 66.8 | 49.7 | 54.0 | 48.8 | 63.0 | 48.7 | 54.2 |
| | | MMD-Agent | **61.6** | **67.8** | **59.3** | **62.1** | **61.5** | **67.7** | **59.1** | **61.0** |

and macro-accuracy as complementary evaluation metrics. Specifically, we construct robust regular expressions to extract key phrases from the long responses for accurate answer matching. Following (Liu et al., 2023b), if a model's response lacks a valid answer, we classify it as a pseudo choice "Z" and consider the response incorrect.

## 5.2 MAIN RESULTS

**Comparison of Different LVLMs.** We present a comprehensive comparison of different LVLMs using the MMFakeBench, detailed in Table 2. Our key findings are summarized as follows:

**1)** *Challenges of MMFakeBench:* The benchmark poses substantial challenges to current models. Notably, GPT-4V, despite its advancement, achieves an F1-score of only 51.0% with the standard prompting. This indicates considerable room for improvement and highlights the rigorous standards of this benchmark.

**2)** *Disparity between Open-source Models and GPT-4V:* Although LLaVA-1.6-34b is the leading open-source model, it achieves an F1-score of just 25.7% with the standard prompting, significantly lower than GPT-4V. This highlights a pronounced disparity in detection capabilities between open-source and proprietary models.

**3)** *Impact of Parameter Quantity:* Comparing models within the same series, such as LLaVA-1.6-Vicuna-7b and LLaVA-1.6-34b, we observe that models with larger parameter counts exhibit better performance. Smaller LVLMs face constraints in instruction-following and high predicted consistency, as detailed in the Appendix A.1.1. These results indicate that 7B parameter models lack sufficient multimodal understanding to effectively combat misinformation.

**4)** *Effectiveness of MMD-Agent:* Due to the limited reasoning capability of small-scale models, we select moderately sized open-source and proprietary models as baselines to compare the proposed MMD-Agent with the standard prompting. MMD-Agent significantly improves the F1-score for both open-source models and GPT-4V. Notably, LLaVA-1.6-34B using MMD-Agent achieves an F1-score of 49.9%, approaching the 51% score of GPT-4V with the standard prompting. This suggests that MMD-Agent can serve as a general framework for future research on the MMFakeBench benchmark.

Table 3: (a) Comparison with single-source detectors for MMFakeBench. (b) Ablation studies on hierarchical (Hier.) decomposition and reasoning knowledge $\mathcal{K}_t = \{\mathcal{R}_t, \mathcal{O}_t\}$ of each sub-task $t$. TVD, VVD, and CCD denote textual veracity distortion, visual veracity distortion, and cross-modal consistency distortion, respectively. Corpus refers to the general datasets used in LVLMs, not tailored for misinformation detection. * denotes the chosen single-source detector applied for mixed detection.

(a)

| Existing Detector | Train Source | Binary Overall↑ | Multiclass Overall↑ |
|---|---|---|---|
| FakingFakeNews* | TVD | 37.8 | - |
| CNNSpot | VVD | 23.8 | - |
| UnivFD | VVD | 28.9 | - |
| LNP* | VVD | 33.0 | - |
| FakeNewsGPT4 | CCD | 41.7 | - |
| HAMMER* | CCD | 43.0 | - |
| Mixed Detection | - | 47.6 | 22.5 |
| LLaVA-1.6-34B | Corpus | 67.2 | 49.9 |
| GPT-4V | Corpus | **74.0** | **61.6** |

(b)

| Hier. | $\mathcal{K}_1$ | $\mathcal{K}_2$ | $\mathcal{K}_3$ | Real↑ | TVD↑ | VVD↑ | CCD↑ | Overall↑ |
|---|---|---|---|---|---|---|---|---|
| | | | | 58.5 | 5.8 | 0.0 | 38.6 | 25.7 |
| ✓ | | | | 45.8 (↓12.7) | 16.5 (↑10.7) | 32.5 (↑32.5) | 49.0 (↑10.4) | 36.0 |
| ✓ | ✓ | | | 46.4 (↑0.6) | 37.6 (↑21.1) | 32.8 (↑0.3) | 47.2 (↓1.8) | 41.0 |
| ✓ | ✓ | ✓ | | 46.8 (↑0.4) | 37.6 (↑0.0) | 61.7 (↑28.9) | 48.6 (↑1.4) | 48.7 |
| ✓ | ✓ | ✓ | ✓ | 51.1 (↑4.3) | 37.6 (↑0.0) | 61.7 (↑0.0) | 49.2 (↑0.6) | 49.9 |
| W/o Wiki Knowledge | | | | 49.7 (↓1.4) | 18.0 (↓19.6) | 61.0 (↓0.7) | 46.3 (↓2.9) | 43.8 |
| W/ Google Knowledge | | | | 50.2 (↓0.9) | 33.4 (↓4.2) | 62.2 (↑0.5) | 48.1 (↓1.1) | 48.5 |

**Comparison with Single-source Detectors.** We compare LLaVA-1.6-34B and GPT-4V utilized in MMD-Agent, with several competitive single-source detectors including FakeingFakeNews (Huang et al., 2023), CNNSpot (Wang et al., 2020), UnivFD (Ojha et al., 2023b), LNP (Liu et al., 2022), FakeNewsGPT4 (Liu et al., 2024c), and HAMMER (Shao et al., 2023). The details of each detector are presented in the Appendix A.3.2. For a fair comparison, in addition to the single-source misinformation detection via existing detectors, we integrate the three most powerful detectors in distinct sources (i.e., FakeingFakeNews, LNP, and HAMMER) to assess the capability of mixed detection. Mixed detection utilizes our proposed hierarchical framework by replacing LVLMs with relevant detectors. The results in Table 3 (a) show that LLaVA-1.6-34B and GPT-4V perform better than single-source detectors for both binary and multiclass classification by a large margin. This demonstrates that LVLMs trained with a large general corpus achieve promising generalization performance in mixed-source MMD and can serve as potential baseline models for future study.

**Results of Human Evaluation.** We conduct a comprehensive user study using a validation set containing 1,000 samples as a question bank. For each questionnaire, 50 samples are randomly selected from this question bank. A total of 80 participants are asked to identify the source of misinformation for each news item. As shown in the first row of Table 2, the results reveal that 62.1% of the items are predicted incorrectly by users, highlighting the dataset's high confusion and the realistic challenge it poses. This level of difficulty and confusion emphasizes the quality and challenge embedded in our dataset, making it an invaluable resource for pushing the boundaries of current understanding and capabilities in multimodal misinformation detection research.

## 5.3 EXPERIMENTAL ANALYSIS

**Ablation Study on Hierarchical Decomposition and Reasoning Knowledge.** We first investigate the effects of instructing LVLMs using only hierarchical decomposition compared to standard prompting. In Table 3 (b), the decomposition method performs better for solving multi-task interference. Additionally, we conduct an ablation study by sequentially generating multi-perspective knowledge for individual sub-tasks. Results in Table 3 (b) show that augmenting decisions with reasoning knowledge outperforms its ablation part, especially for checking content veracity. We further conduct ablations on external knowledge by removing it or integrating an alternative source (i.e., Google Knowledge Graph). Results show that models using external knowledge outperform those relying solely on internal reasoning, highlighting its critical role in validating textual veracity.

Table 4: Performance (F1 score (%)) of models on different sources of misinformation.

| Model | Real↑ | TVD↑ | VVD↑ | CCD↑ | Overall↑ |
|---|---|---|---|---|---|
| VILA-13B | 32.4 | 13.4 | 4.3 | 37.6 | 21.9 |
| InstructBLIP-13B | 41.9 | 18.8 | 19.6 | 23.8 | 26.0 |
| BLIP2-FLAN-T5-XXL | 41.5 | 39.2 | 13.1 | 32.6 | 31.6 |
| LLaVA-1.6-34B | 51.1 | 37.6 | 61.7 | 49.2 | 49.9 |
| GPT-4V | **65.3** | **67.2** | **57.3** | **56.5** | **61.6** |

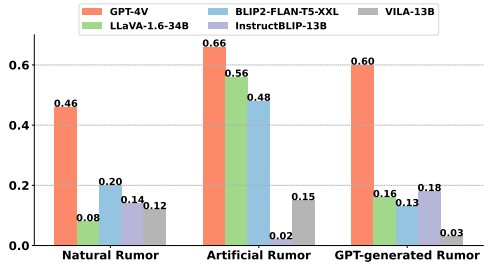

Figure 5: Performance (detection success rate) of models on different types of rumor.

**Analysis of Misinformation Sources and Types.** We compare the F1 scores of various LVLMs across misinformation sources in Table 4. Across all sources, the majority of open-source models perform worse than GPT-4V by a huge margin, particularly in terms of textual veracity distortion. This indicates that open-source models are considerably challenged by textual veracity distortion. Within textual veracity distortion, we further report the detection accuracy of selected models across three types of textual rumors. Fig. 5 shows that open-source models typically perform better in the artificial rumor. This might be attributed to the fact that artificial rumors are constructed by applying specific rules, which provide identifiable points of falsification enabling verification through external knowledge. In contrast, natural or GPT-generated rumors often leverage ambiguous language that makes detection challenging. Analysis of other misinformation sources is detailed in Appendix A.1.3.

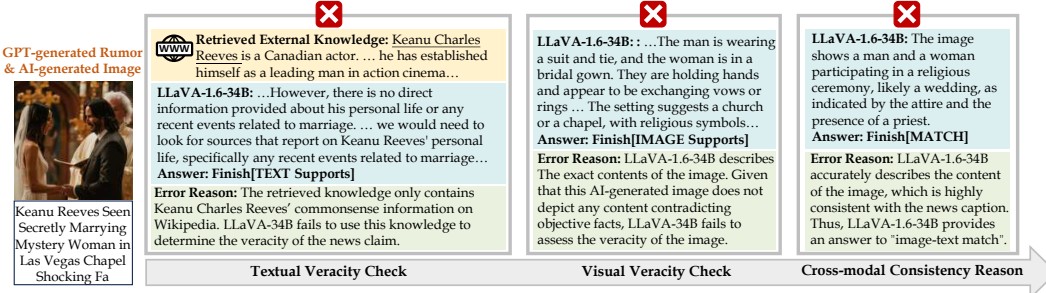

Figure 6: One of the most harmful examples involves a GPT-generated rumor supported by an AI-generated image, which is challenging for LLaVA-1.6-34B. More examples can be found in the Appendix A.1.7.

**Error Analysis.** A fundamental root cause of textual veracity checking errors in the LLaVA-1.6-34B is the lack of useful external knowledge. This deficiency is exemplified in Fig. 6, where the knowledge contains only commonsense information but fails to provide relevant events. Moreover, this AI-generated image in the image-text pair exhibits high fidelity and strong coherence, thus evading detection in visual veracity and cross-modal consistency. These instances underscore the dangers of using collaborative generative modes to automatically generate multimodal misinformation.

## 6 CONCLUSION

In this paper, we introduce MMFakeBench, the first comprehensive benchmark for detecting mixed-source multimodal misinformation. MMFakeBench contains three primary misinformation categories along with 12 sub-categories of forgery types. We conduct comprehensive evaluations of 6 prevalent detection methods and 15 LVLMs on the MMFakeBench dataset. Furthermore, we propose an innovative unified framework and perform extensive experiments to demonstrate its effectiveness.

## ETHICS STATEMENT AND LIMITATIONS

This paper contains examples of harmful texts or images, raising concerns about the manipulation of public safety. To reduce its social impact, we have implemented several safeguards: (1) Content Safeguards. Rigorous review sensitive content in data generation such as those related to politics and race etc. (2) Disable data generation code access. We open-source datasets and detection codes but do not release data generation codes for safety. (3) Data Access Restrictions. Data access is restricted to verified researchers by following a binding usage agreement. (4) Public Feedback Mechanism. We will offer a public feedback channel for ethical concerns and continuous dataset improvement. The licenses for the datasets contributed in this work are discussed in Appendix A.5.

While our MMFakeBench marks a critical advancement in mixed-source multimodal misinformation detection, it is important to recognize certain limitations. Our proposed framework utilizes external knowledge retrieved from the Wikipedia API. While the integration of such external knowledge has enhanced the performance of our baseline models, it may not always provide useful information for particularly challenging natural rumors and GPT-generated rumors. Future research should explore the use of a more advanced retrieval augmentation generation (RAG), which could lead to further performance improvements.

ACKNOWLEDGMENTS

This research is sponsored by Beijing Nova Program (Grant No. 20230484488, Z211100002121106), National Natural Science Foundation of China (Grant No. 62306041, No.62176025, No. U21B2045), Youth Innovation Promotion Association CAS(Grant No.2022132), Beijing Nova Program(20230484276), Beijing Natural Science Foundation (4252054).

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

# A APPENDIX

This appendix contains additional details for the ICLR 2025 submission, titled "MMFakeBench: A Mixed-Source Multimodal Misinformation Detection Benchmark for LVLMs". The appendix is organized as follows:

- §A.1 Additional Experimental Result.

  - §A.1.1 Instruction Following and Prediction Consistency.
  - §A.1.2 Evaluation Results for Binary Classification.
  - §A.1.3 Analysis of More Misinformation Types.
  - §A.1.4 Comparision of Existing Datasets.
  - §A.1.5 Efficiency of MMD-Agent.
  - §A.1.6 Fine-tuning on Existing Detectors.
  - §A.1.7 More Error Analysis.
  - §A.1.8 Interpretable visualization.

- §A.2 Benchmark Analysis.

- §A.3 Implementation Details.

  - §A.3.1 LVLMs Configuration Details.
  - §A.3.2 Existing Single-source Detectors Details.

- §A.4 Instruct Prompts for ChatGPT.

- §A.5 Dataset Licenses.

- §A.6 More Visualization Examples.

## A.1 ADDITIONAL EXPERIMENTAL RESULTS

Table 5: Statistics of Instruction following capabilities and predicted consistency tendency of LVLMs.

| Model | Match | Consist. | Model | Match | Consist. | Model | Match | Consist. |
|---|---|---|---|---|---|---|---|---|
| **LVLMs with 7B Parameter** | | | | | | | | |
| Otter-Image | 9.8 | 100 | Qwen-VL | 88.6 | 92.9 | mPLUG-Owl2 | 100 | 96.6 |
| MiniGPT4 | 100 | 88.6 | VILA | 100 | 100 | BLIP2 | 100 | 93.6 |
| InstructBLIP | 25.7 | 96.11 | PandaGPT | 100 | 98.9 | LLaVA-1.6 | 100 | 76.9 |
| **LVLMs with 13B Parameter** | | | | | | | | |
| VILA | 100 | 100 | InstructBLIP | 99.9 | 91.0 | BLIP2-FlanT5-XXL | 100 | 49.4 |
| **LVLMs with 34B Parameter** | | | | | | | | |
| LLaVA-1.6 | 100 | 52.6 | - | - | - | - | - | - |
| **Proprietary LVLMs** | | | | | | | | |
| GPT-4V | 99.9 | 36.5 | - | - | - | - | - | - |

### A.1.1 INSTRUCTION FOLLOWING AND PREDICTION CONSISTENCY.

We evaluated the instruction following capabilities and prediction consistency to further study the multimodal understanding of LVLMs on mixed-source multimodal misinformation detection (MMD). We report the success rate in heuristic matching (Match) with regular expressions and prediction consistency rate (Consist.). The results are shown in the Table 5. Among all LVLMs, small-scale models like Otter-Image and InstructBLIP, achieve the lower matching success rate. While there exist small-scale LVLMs that perfectly follow the format of the regular expressions and achieve high success rates (>99%) in matching, most small-scale models exhibit high predicted consistency rates. This indicates small-scale models may prefer to predict a certain category answer among all given choices. Additionally, the leading open-source model, LLaVA-1.6-34B, and the proprietary model GPT-4V demonstrate superior instruction-following capabilities and lower prediction consistency. This indicates their significant potential in addressing mixed-source MMD, positioning them as valuable baseline models.

Table 6: Binary overall results of different models on the MM-FakeBench validation and test set with the comparison of standard prompting (Standard) and proposed MMD-Agent framework.

| Model | Language | Prompt | Validation (1000) | | | | Test (10000) | | | |
|---|---|---|---|---|---|---|---|---|---|---|
| Name | Model | Method | F1 | Precision | Recall | ACC | F1 | Precision | Recall | ACC |
| *Human Evaluation* | | | 54.9 | 56.6 | 57.8 | 56.8 | - | - | - | - |
| **LVLMs with 7B Parameter** | | | | | | | | | | |
| Otter-Image | MPT-7B | Standard | 7.9 | 4.1 | 4.5 | 7.9 | 8.6 | 32.4 | 5.0 | 8.6 |
| MiniGPT4 | Vicuna-7B | Standard | 40.4 | 38.2 | 45.7 | 63.1 | 41.7 | 41.0 | 47.4 | 65.2 |
| InstructBLIP | Vicuna-7B | Standard | 14.7 | 30.8 | 13.2 | 8.1 | 16.1 | 40.5 | 14.2 | 8.8 |
| Qwen-VL | Qwen-7B | Standard | 43.6 | 50.6 | 44.9 | 60.3 | 44.0 | 51.6 | 45.2 | 60.5 |
| VILA | LLaMA2-7B | Standard | 41.2 | 35.0 | 50.0 | 70.0 | 41.2 | 35.0 | 50.0 | 70.0 |
| PandaGPT | Vicuna-7B | Standard | 24.6 | 60.6 | 50.5 | 30.9 | 24.1 | 61.7 | 50.4 | 30.6 |
| mPLUG-Owl2 | LLaMA2-7B | Standard | 47.2 | 64.9 | 52.3 | 70.6 | 48.7 | 71.1 | 53.3 | 71.4 |
| BLIP2 | FlanT5-XL | Standard | 41.2 | 35.0 | 50.0 | 70.0 | 41.2 | 35.0 | 50.0 | 70.0 |
| LLaVA-1.6 | Vicuna-7B | Standard | 48.1 | 48.2 | 48.5 | 59.5 | 52.5 | 53.0 | 52.6 | 62.5 |
| **LVLMs with 13B Parameter** | | | | | | | | | | |
| VILA | LLaMA2-13B | Standard | 41.1 | 35.0 | 50.0 | 70.0 | 41.1 | 35.0 | 50.0 | 70.0 |
| | | MMD-Agent | **56.5** | **62.2** | **56.9** | **70.3** | **56.6** | **64.3** | **57.2** | **71.2** |
| InstructBLIP | Vicuna-13B | Standard | 41.1 | 35.0 | 49.9 | **69.9** | 41.1 | 35.0 | 49.9 | **69.8** |
| | | MMD-Agent | **51.3** | **53.4** | **54.0** | 53.1 | **47.9** | **50.1** | **50.1** | 49.9 |
| BLIP2 | FlanT5-XXL | Standard | 31.6 | **63.4** | 53.6 | 35.5 | 30.6 | **64.9** | 53.4 | 34.9 |
| | | MMD-Agent | **51.5** | 53.4 | **54.0** | **53.6** | **51.8** | 54.0 | **54.7** | **53.5** |
| LLaVA-1.6 | Vicuna-13B | Standard | 41.1 | 35.0 | 50.0 | 69.7 | 42.3 | 57.3 | 50.1 | 69.5 |
| | | MMD-Agent | **51.8** | **66.7** | **54.6** | **71.4** | **50.2** | **67.3** | **53.9** | **71.3** |
| **LVLMs with 34B Parameter** | | | | | | | | | | |
| LLaVA-1.6 | Nous-Hermes-2 | Standard | 62.9 | 67.1 | **70.0** | 63.4 | 64.3 | 68.8 | **71.7** | 64.8 |
| | -Yi-34B | MMD-Agent | **67.2** | **70.4** | 66.0 | **75.1** | **68.1** | **71.1** | 67.0 | **75.6** |
| **Proprietary LVLMs** | | | | | | | | | | |
| GPT-4V | ChatGPT | Standard | 72.3 | 72.1 | 72.8 | 75.6 | **74.2** | **73.5** | **76.9** | **76.4** |
| | | MMD-Agent | **74.0** | **73.4** | **75.5** | **76.8** | 72.8 | 72.4 | 75.4 | 75.0 |

### A.1.2   EVALUATION RESULTS FOR BINARY CLASSIFICATION.

In addition to multi-class classification, we also provide binary classification performance to assess the overall detection capability of baseline models in mixed-source MMD. Based on the 4 categories in the mixed-source MMD settings, we develop binary evaluation metrics via mapping techniques. Specifically, we standardize the assignment of labels denoting "textual veracity distortion", "visual veracity distortion", and "cross-modal consistency distortion" to the classification of "Fake" while reserving the label "True" to denote real data. Similar to the multi-classification evaluation, we adopt the widely used F1 score. In addition to the F1 score, we also use precision, recall, and accuracy as supplementary evaluation metrics. The specific evaluation results are shown in Table 6. From the results, we make the following observations:

• Current models including open-source models and GPT-4V are challenged by the MMFakeBench dataset in binary classification detection. Despite being an advanced modal, GPT-4V attains a mere F1-score of 72.3% using the standard prompting on the validation set.

• The proposed framework MMD-Agent yields substantial improvement on recent LVLMs, especially on open-source models. For instance, for the BLIP2-FlanT5-XXL model, MMD-Agent achieves a 19.9% increase in F1 score on the validation set. This may be credited to the effective integration of reasoning, actions, and tool use in enhancing multimodal understanding in mixed-source MMD.

### A.1.3   ANALYSIS OF MORE MISINFORMATION TYPES.

Within visual veracity distortion, we report the detection accuracy of selected models across two types of fact-conflicting images. Fig. 7 (a) shows the challenging nature of both types of such fact-conflicting images for existing models. Notably, even the advanced GPT-4V achieves a detection success rate of less than 50% on both types. Additionally, we present an analysis of the detection accuracy of selected models across different types of image-text inconsistency. As shown in Fig. 7 (b),

edited inconsistency emerges as a more substantial challenge compared to repurposed inconsistency. This finding suggests that editing methods introduce minor alterations to images or text, necessitating enhanced multimodal reasoning capabilities.

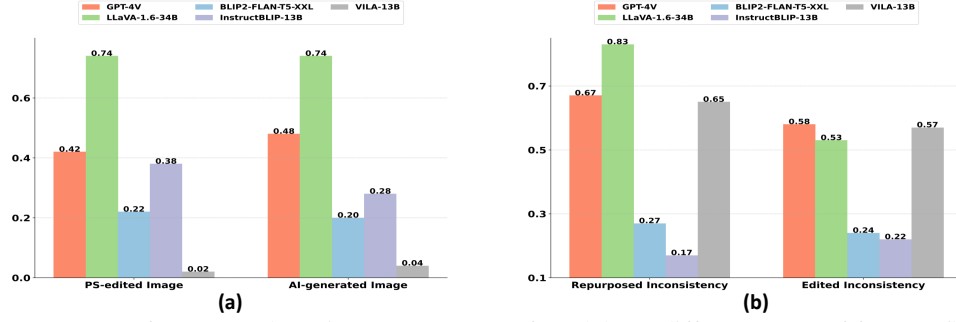

Figure 7: (a) Performance (detection success rate) of models on different types of fact-conflicting images. (b) Performance of models on different types of inconsistent image-text pairs.

Table 7: Performance (F1 score) comparison of selected models on the proposed benchmark and existing benchmarks (datasets).

| Model Name | NewsClipings (Binary) | NewsClipings (Multiclass) | Fakeddit (Binary) | Fakeddit (Multiclass) | MMFakeBench (Binary) | MMFakeBench (Multiclass) |
|---|---|---|---|---|---|---|
| Large Vision Language Models | | | | | | |
| BLIP2-FLanT5-XXL | 33.6 | - | 40.5 | | 31.6 | 16.7 |
| LLaVA-1.6-34B | 65.6 | - | 71.4 | | 62.9 | 25.7 |
| Multimodal Specialized Detectors | | | | | | |
| FKA-Owl | 52.0 | - | 55.0 | | 41.7 | - |
| HAMMER | 55.0 | - | 52.6 | | 43 | - |

### A.1.4 COMPARISION OF EXISTING DATASETS

We conduct an experiment to compare the performance of selected models on the proposed benchmark against existing benchmarks (datasets) in Table 7. We employ four models: two powerful open-source LVLM models, LLaVA-1.6-34B and BLIP2-FLANT5-XXL with two open-source multimodal specialized detectors, HAMMER (Shao et al., 2023) and FKA-Owl (Liu et al., 2024c). All these models are evaluated on two widely-used multimodal misinformation datasets, NewsClippings (Luo et al., 2021) and Fakeddit (Nakamura et al., 2020). The results indicate that our MMFakeBench poses a greater challenge in binary classification compared to the other two datasets. More importantly, the primary challenge of our benchmark lies in its capacity to perform fine-grained assessments of misinformation sources in scenarios where multiple forgery types coexist. This mirrors the complexity of real-world environments, where misinformation stems from diverse, overlapping sources. This capability is essential for addressing real-world challenges and underscores the importance of MMFakeBench in advancing multimodal misinformation detection.

### A.1.5 EFFICIENCY OF MMD-AGENT

In Table 8, we compare the inference time and computational resource usage of MMD-Agent with standard prompting methods. Overall, while MMD-Agent shows an increase in inference time, it does not lead to additional GPU memory consumption and significantly enhances

Table 8: Efficiency of MMD-Agent on LLaVA-1.6-34B.

| Metric | Standard | MMD-Agent |
|---|---|---|
| Average Inference Time (s) | 5.97s | 49.04s |
| Memory (GB) | 82G | 82G |
| Performance (F1 score) | 25.7 | 49.9 |

both the performance and interpretability of multimodal misinformation detection. The experimental results are shown in the table, and the detailed analysis is as follows: (1) MMD-Agent introduces higher inference time compared to standard prompting methods, primarily due to the additional reasoning steps, such as extracting key entities from the text. (2) MMD-Agent does not lead to additional GPU memory consumption. This is because the Agent method does not affect the model parameter deployment and dataset storage.

While the increased inference time is a consideration, the substantial gains brought by the MMD-Agent should not be overlooked. On the one hand, it greatly enhances the detection performance, with the F1 score improving from 25.7 to 49.9. On the other hand, MMD-Agent offers stronger interpretability because it offers a detailed analysis of the misinformation rather than only providing classification labels as used in standard prompting methods. Specifically, as shown in Fig. 4, the content of the intermediate reasoning traces such as Thought 2, Thought 3, and Thought 4 accurately analyzes the specific reasons for misinformation from different sources.

Table 9: (a) Results of the specialized detector, HAMMER, fine-tuning on the MMFakeBench dataset. (b) Results of the specialized detector, FKA-Owl, fine-tuning on the MMFakeBench dataset.

(a) Fine-tune on HAMMER Model.

| Model | DGM4 | MMFakeBench |
| --- | --- | --- |
| HAMMER before fine-tuning | 83.2 | 44.1 |
| HAMMER after fine-tuning (10) | 78.7 | 46.8 |
| HAMMER after fine-tuning (100) | 70.1 | 60.8 |
| HAMMER after fine-tuning (1000) | 63.2 | 72.4 |

(b) Fine-tune on FKA-Owl Model.

| Model | DGM4 | MMFakeBench |
| --- | --- | --- |
| FKA-Owl before fine-tuning | 78.5 | 44.6 |
| FKA-Owl after fine-tuning (10) | 78.3 | 46.9 |
| FKA-Owl after fine-tuning (100) | 66.4 | 61.1 |
| FKA-Owl after fine-tuning (1000) | 64.8 | 76.2 |

## A.1.6 FINE-TUNING ON EXISTING DETECTORS

We conducted experiments in Table 9 to investigate the adaptation of existing models to the MM-FakeBench dataset. Specifically, we selected two existing powerful specialized detectors, HAMMER Shao et al. (2023) and LVLM-based detector, FKA-Owl Liu et al. (2024c). Then we fine-tuned them incrementally with 10, 100, and 1000 examples. We reported the results on both the DGM4 Shao et al. (2023) dataset (used for training and evaluating HAMMER and FKA-Owl) and our proposed dataset. The results indicate:

**1) Tuning-based models can be hard to generalize to unseen forgery data.** For off-the-shelf dedicated detectors without fine-tuning, their performance on the proposed benchmark dataset is notably poor. With the rapid development of generative models, new forgery techniques and synthesized data continue to emerge. Models trained on limited samples face significant challenges in generalizing to unseen types of forgery data, highlighting a critical issue in the field of fake detection Liu et al. (2024c); Ojha et al. (2023b).

**2) Catastrophic forgetting issues are inevitable.** Fine-tuning on new data improves performance on MMFakeBench but simultaneously leads to a decline in performance on the original dataset. This degradation underscores the phenomenon of catastrophic forgetting, where previously acquired knowledge, such as that from the DGM4 dataset, is progressively lost.

## A.1.7 MORE ERROR ANALYSIS

In Fig. 8, We have presented more error analysis that span four models of different scales (i.e., GPT-4V, LLaVA-1.6-34B, BLIP2-FLAN-T5-XXL, and VILA-13B) and on three distinct forgery sources of textual veracity distortions, visual veracity distortion and cross-modal consistency distortion. Based on these cases, we provide a deep analysis of the shortcomings of both largely and moderately sized models when encountering different sources of multimodal misinformation.

**1) For cases of textual veracity distortion**, our analysis is summarized as follows:

• Reliance on external knowledge. While Largely sized models like LLaVA-1.6-34B and GPT-4V exhibit strong reasoning capabilities, they are challenging to infer the factual correctness of a statement without access to a broader context. For instance, when faced with factually incorrect statements that appear linguistically accurate, such as the GPT-generated rumor, all these four models lack the inherent capability to question the information without retrieving corroborative evidence.

**2) For cases of visual veracity distortion**, our analysis is summarized as follows:

• Limited Sensitivity to Abnormal Physical Features. Models including GPT-4V, LLaVA-1.6-34B, BLIP2-FLAN-T5-XXL, and VILA-13B, face challenges in discerning abnormal physical characteristics. For instance, in the PS-edited examples, these four models fail to detect subtle yet unrealistic manipulations, such as swollen necks or distorted facial features in images of Donald Trump. Instead, they are frequently distracted by more prominent visual elements, such as facial expressions or gestures, resulting in misjudgments when identifying these manipulations.

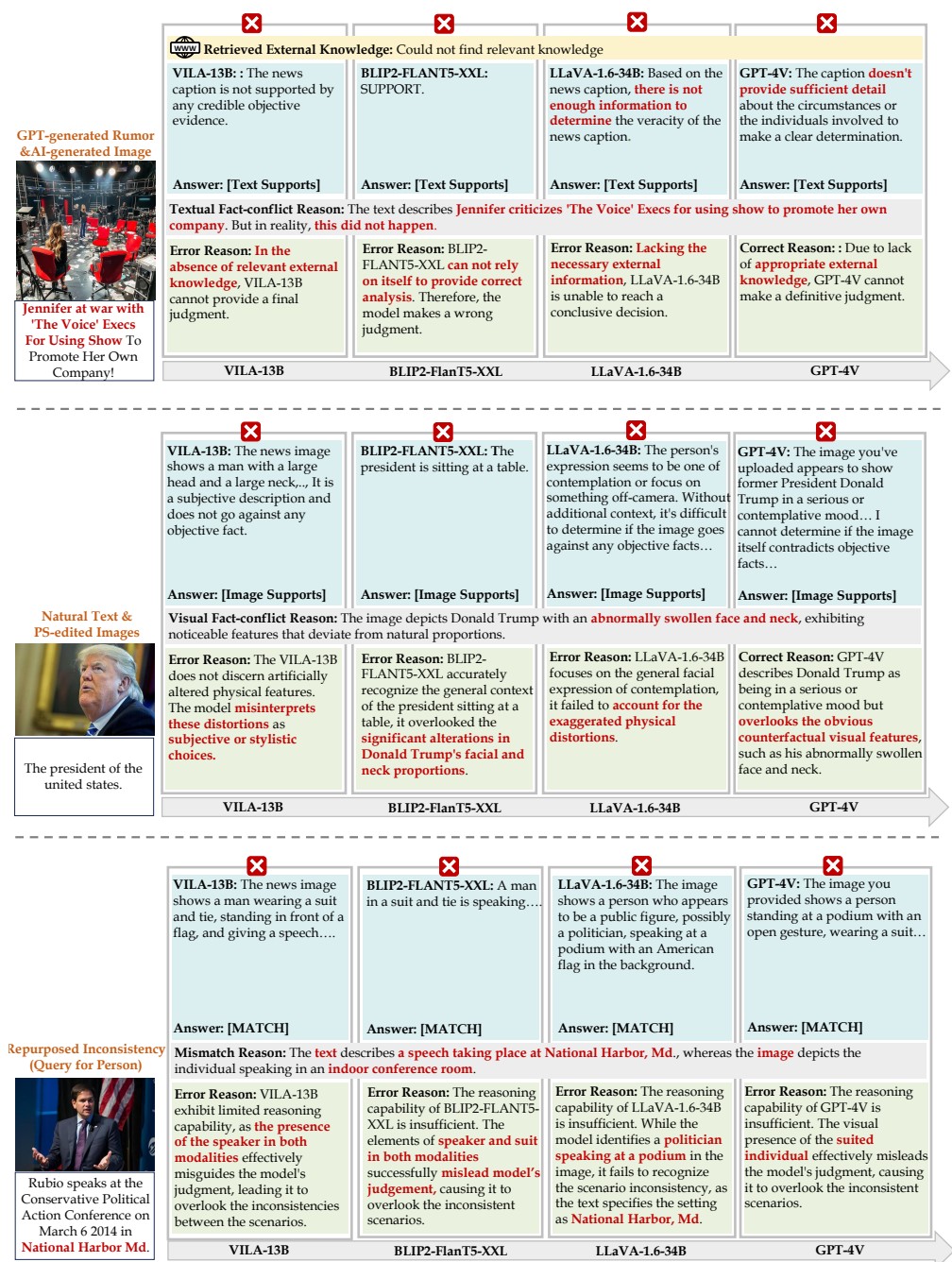

Figure 8: More Error analysis from three different forgery sources.

**3) For cases of cross-modal consistency distortion**, our analysis is summarized as follows:

• Distraction by global consistent semantics. When confronted with scenarios where images and text exhibit only subtle inconsistencies while other aspects remain largely consistent, large-scale models such as LLaVA-1.6-34B and GPT-4V often struggle to detect these discrepancies. These models can be distracted by the dominant presence of consistent content, which obscures the subtle mismatches critical for accurate misinformation detection. For instance, in the case of repurposed inconsistency, all four models focus on global semantics, such as a person delivering a speech, which diverts attention from deeper, subtle inconsistencies in the scenario.

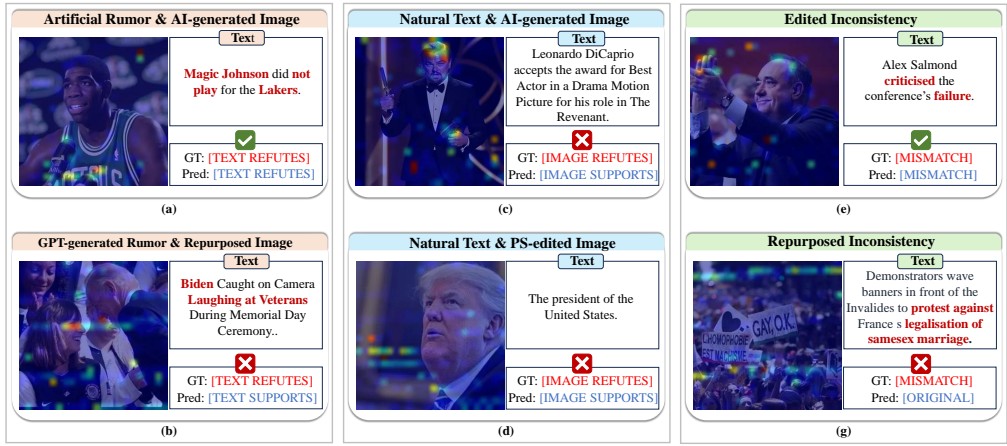

Figure 9: Illustration of relevancy maps to showcase interpretability for the predicted output of the LLaVA-13B model.

### A.1.8 INTERPRETABLE VISUALIZATION

We have incorporated the recent LVLM-interpret Stan et al. (2024) as a visualization tool to provide a more transparent understanding of the model's decision-making process. This method adapts the calculation of relevancy maps to LVLM, thus providing a detailed representation of the regions of the image most relevant to each generated token. Specifically, as shown in Fig. 9 (e) for cross-modal consistency distortion, the relevancy map focuses on the clapping action of the characters in the image. Clapping typically signifies recognition or approval of an event, conveying semantics that differ from the meaning of the associated text. This observation aligns with the model's detection output, providing intuitive explanations for the identified inconsistency. In contrast, Fig. 9 (g) shows that the model predominantly focuses on the person depicted in the image while neglecting the slogan displayed at its center. This oversight prevents the model from identifying the semantic inconsistencies between the textual and visual modalities, resulting in a failure to capture the underlying mismatch.

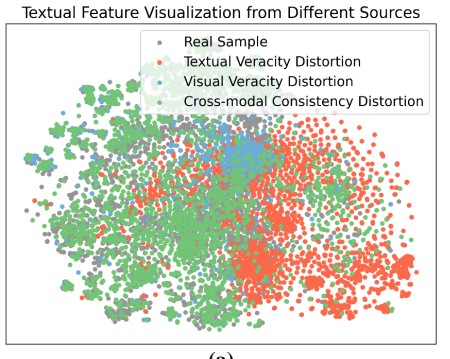
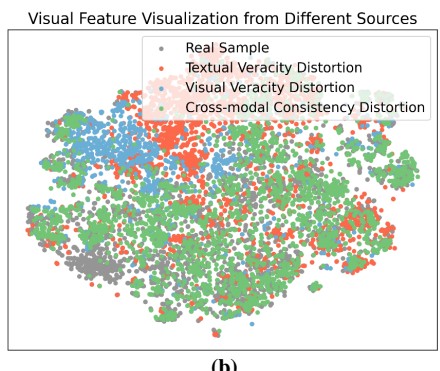

Figure 10: a) TSNE visualization of textual features from different sources. (b) TSNE visualization of visual features from different sources.

### A.2 BENCHMARK ANALYSIS

In Table 10, We present benchmark analysis based on count statistics (i.e., the average number of words) and diversity (i.e., feature distribution and word frequency) and semantic similarity metrics for different forgery sources. The results indicate: 1) The average number of words is over 10 words, which suggests that the samples are sufficiently informative. 2) We have provided an analysis of the dataset's diversity through t-SNE visualizations of textual and visual features presented in Fig. 10, as well as

Table 10: Benchmark statistics analysis.

| Metric | Real Unit | Fake Unit |
|---|---|---|
| Avg. Number of Words | 15.8 | 12.6 |
| Word Frequency Entropy | 10.9 | 11.6 |
| Semantic Similarity (Text-Text) | 0.41 | 0.47 |
| Semantic Similarity (Image-Image) | 0.39 | 0.41 |

through the entropy of word frequencies detailed in Table 10. This feature visualization provides a clear depiction of the dataset's diversity where each forgery sources are distinctly dispersed across the feature space. Moreover, we calculate the overall entropy of word frequency, where the entropy for real units ranges from 0 to 15.5 (calculated by $\log_2 (3000 \times 15.8)$), 0 to 16.4 (calculated by $\log_2 (7000 \times 12.6)$) for fake units where this dataset's entropy reached 10.9 and 11.6, demonstrating significant diversity. (3) Additionally, we computed the pairwise semantic similarity between samples, with results showing an average similarity below 0.5, further confirming the dataset's rich diversity.

## A.3 IMPLEMENTATION DETAILS

### A.3.1 LVLMs CONFIGURATION DETAILS

**Model Version.** As for ChatGPT model, we use GPT-3.5 (gpt-3.5-turbo) or GPT-4 (gpt-4-vision-preview) as generators or detectors. As for text-to-image models, we use DALLE (DALLE-E3), Stable-Diffusion (Stable Diffusion XL), and Midjourney (Midjourney V6).

**Inference Hyperparameters.** To achieve the justified evaluation, we have set the sampling hyperparameter of the off-the-shelf LVLMs, "do_sample = False" or "Temperature = 0", to guarantee consistency in the predicted outputs. We adopt the default setting of other hyperparameters such as "max_new_tokens = 512".

### A.3.2 EXISTING SINGLE-SOURCE DETECTORS DETAILS

**FakeingFakeNews.** The FakingFakeNews Huang et al. (2023) is designed for the detection of textual fake news, particularly for those natural human-written misinformation. It proposes an innovative approach for generating training instances, leveraging established styles and strategies commonly employed in human-authored propaganda. FakingFakeNews employs the ROBERTA Liu et al. (2019) model as the backbone and trains it on its own proposed PROPANEWS dataset. In our experiments, we utilized the default configuration of the ROBERTA detector provided within the FakingFakeNews framework, retaining its default hyperparameters.

**CNNSpot.** CNNSpot Wang et al. (2020) is an artificial image detector designed specifically for identifying images produced by generative models. It employs the ResNet-50 model as the classifier backbone. Notably, CNNSpot recognizes that data augmentation, including JPEG compression and Gaussian blur, can enhance the generalization capabilities of the detector. In our study, we utilize the pre-trained CNNSpot model with default hyperparameters to perform the detection of visual veracity distortion.

**UnivFD.** UvinFD Ojha et al. (2023a) is a general-purpose fake image detector that uses a feature space not explicitly trained to distinguish between real from fake images. When given access to the feature space of a pre-trained vision-language model, UvinFD employs the nearest neighbor to identify fake images originating from various sources. The utilization of the large pre-trained model results in a smooth decision boundary, thereby enhancing the generalization capability of the detector. In our work, we use the pre-trained detector of UnivFD with default hyperparameters to conduct the visual veracity distortion detection task.

**LNP.** LNP Liu et al. (2022) utilizes a well-trained denoising model to extract noise patterns from spatial images. Subsequently, it discerns fake images by analyzing the frequency domain of these noise patterns. Additionally, LNP employs the ResNet-50 model as the classifier backbone. In our study, we utilize the pre-trained LNP detector with default hyperparameters to conduct the visual veracity distortion detection task.

**HAMMER.** HAMMER Shao et al. (2023) is a multimodal detector designed to identify multimedia manipulation. It is built upon the pre-trained vision-language model, ALBEF Li et al. (2021) which comprises two unimodal encoders and a multimodal Aggregator. To accomplish the multimodal manipulation detection task, HAMMER employs hierarchical manipulation reasoning consisting of shallow and deep manipulation reasoning. Shallow manipulation reasoning involves semantic alignment between image and text embeddings, while deep manipulation reasoning performs deep

cross-modal fusion for forgery detection. In our paper, we employ the off-the-shelf HAMMER detector with default hyperparameters to detect cross-modal consistency distortion.

**FakeNewsGPT4.** FakeNewsGPT4 Liu et al. (2024c) is developed based on large vision language models (LVLMs) to detect multimodal fake news. It identifies two types of forgery-specific knowledge: semantic correlation and artifact tract, and augments LVLMs with these two knowledge. Specifically, it extracts semantic correlations using a multi-level cross-modal reasoning module and comprehends unimodal localized details through a dual-branch fine-grained verification model. In our study, we employed the off-the-shelf FakeNewsGPT4 with default hyperparameters to accomplish the cross-modal consistency distortion detection task.

**Mixed Detection** We combine the three most powerful models on each single-source detection task (i.e., FakingFakeNews for textual veracity distortion, LNP for visual veracity distortion, and HAMMER for cross-modal consistency distortion) to perform mixed detection. Specifically, utilizing our proposed hierarchical decomposition framework, we sequentially assess textual veracity, visual veracity, and cross-modal consistency with three single-source detectors and assign corresponding multi-class labels.

## A.4 INSTRUCT PROMPT FOR CHATGPT

The construction of MMFakeBench employs the advanced ChatGPT to assist us in generating textual rumors, expanding detailed descriptions, and generating fact-conflicting descriptions. The specific prompts provided in this work are summarized as follows.

**Instruct Prompts to Ask ChatGPT to Generate Textual Rumors.** Fig. 11 illustrates the prompt utilized for asking ChatGPT to generate textual rumors with different prompt methods. These methods include arbitrary generation, rewriting generation, and information manipulation.

### (1) Arbitrary Generation Prompt

> Please write a piece of misinformation title. The domain should be one of gossip, science, health and politics. The time period should be within the past ten years. The type should be fake news/rumors/misleading claims. Avoid answering words like fake, rumor, confusion, disbelief, misinformation, etc.
> #query
> Misinformation title is:

### (2) Rewriting Generation Prompt

> Given a sentence, please write a piece of misinformation title . The content should be the same. The writing style should be serious, informative and convincing. Avoid answering words like fake, rumor, confusion, disbelief, misinformation, etc.
> #query
> Sentence:___________.
> Misinformation title is:

### (3) Information Manipulation Prompt

> Given a true claim, please write a piece of misinformation. It should be long enough, convincing and detailed. The error type should be fake news/rumors/misleading claims. Avoid answering words like fake, rumor, confusion, disbelief, misinformation, etc.
> #query
> The true claim is:___________.
> The answer is:

Figure 11: Prompt used to ask ChatGPT to generate textual rumors with arbitrary generation, rewriting generation, and information manipulation methods.

**Instruct Prompts to ask ChatGPT to Expand Detailed Descriptions.** Fig. 12 illustrates the prompt utilized for asking ChatGPT to expand detailed descriptions for textual rumors. Based on

Given a news rumor that is not consistent with reality, I need you to imagine the scene in which the rumor occurred and provide details about the characters, famous buildings, physical objects, etc. in the scene. Avoid sentences that state opinions and only describe physical objects. Avoid words like false, rumor, confusion, disbelief, misinformation, etc.

# in-context examples
Rumor: Peking University is in Thailand.
The answer is: A Thai university in the real world, with the Thai flag flying above, including the landmark building of Peking University, the gate of Peking University, a plaque with the name of Peking University, the Boya Tower of Peking University, and Weiming Lake of Peking University.

Rumor: Carlos Santana is a US president.
The answer is: This a realistic photo of Carlos Santana in the white house in the real world. Carlos Santana stands at a podium adorned with the presidential seal. Behind him, an American flag hangs proudly. A row of microphones and a table are set before him, as cameras flash around the room.

Rumor: The Chrysler Building has yet to be surpassed in height.
The answer is: This is a realistic photo of the center of New York City in the real world. The Chrysler Building stands tall. The slope of the Chrysler Building forms a crown. There are many high-rise buildings on the side.

#query
Rumor: ___________.
The answer is:

Figure 12: Prompt used to ask ChatGPT to expand detailed descriptions of rumors.

Imagine you as a science fiction writer. Given a true claim, please write a piece of misinformation. The error type should be contrary to objective facts, that is, objects or events that do not exist in the real scene, etc. Avoid answering words like fake, rumor, confusion, disbelief, misinformation, etc.
# in-context examples
The true claim is: People are standing on top of a snowy mountain.
The answer is: There are angels with wings welcoming these people.

The true claim is: A person sailing in the air on a snow board.
The answer is: A person is snowboarding in the clouds.

The true claim is: A man wearing a blue tie with the ten commandments on it.
The answer is: This man is holding a huge fireball in his hand.
#query
The true claim is: ___________.
The answer is:

Figure 13: Prompt used to ask ChatGPT to generate fact-conflicting descriptions.

the responses, we prompt the stage-of-art diffusion generators to generate realistic and relevant supporting images.

**Instruct Prompts to ask ChatGPT to Generate Fact-conflicting Descriptions.** Fig. 13 illustrates the prompt utilized for asking ChatGPT to generate fact-conflicting descriptions. Then, we combine these descriptions with original captions as prompts in the Midjourney V6 model Midjourney (2022) to create corresponding images with additional fact-conflicting information.

## A.5 DATASET LICENSES

The licenses of the existing datasets used in this work is as follows:

- **FakeNewsNet**: free to use by all.
- **FEVER**: Apache License 2.0.
- **Fakeddit**: free to use by all.

- **NewsClipings**: free to use by all.
- **DGM4**: S-Lab License 1.0
- **COCO-Counterfactuals**: Attribution 4.0 International (CC BY 4.0).
- **VisualNews**: free to use by all.
- **MSCOCO**: free to use by all.

All datasets provided in this work are licensed under the Attribution Non-Commercial ShareAlike 4.0 International (CC BY-NC-SA 4.0) license. We chose this license because some of the original datasets have this license and we provide our datasets with the same level of access.

### A.6 More Visualization Examples

We have provided more visualization examples of multimodal misinformation from different sources in Fig. 14, Fig. 15 and Fig. 16.

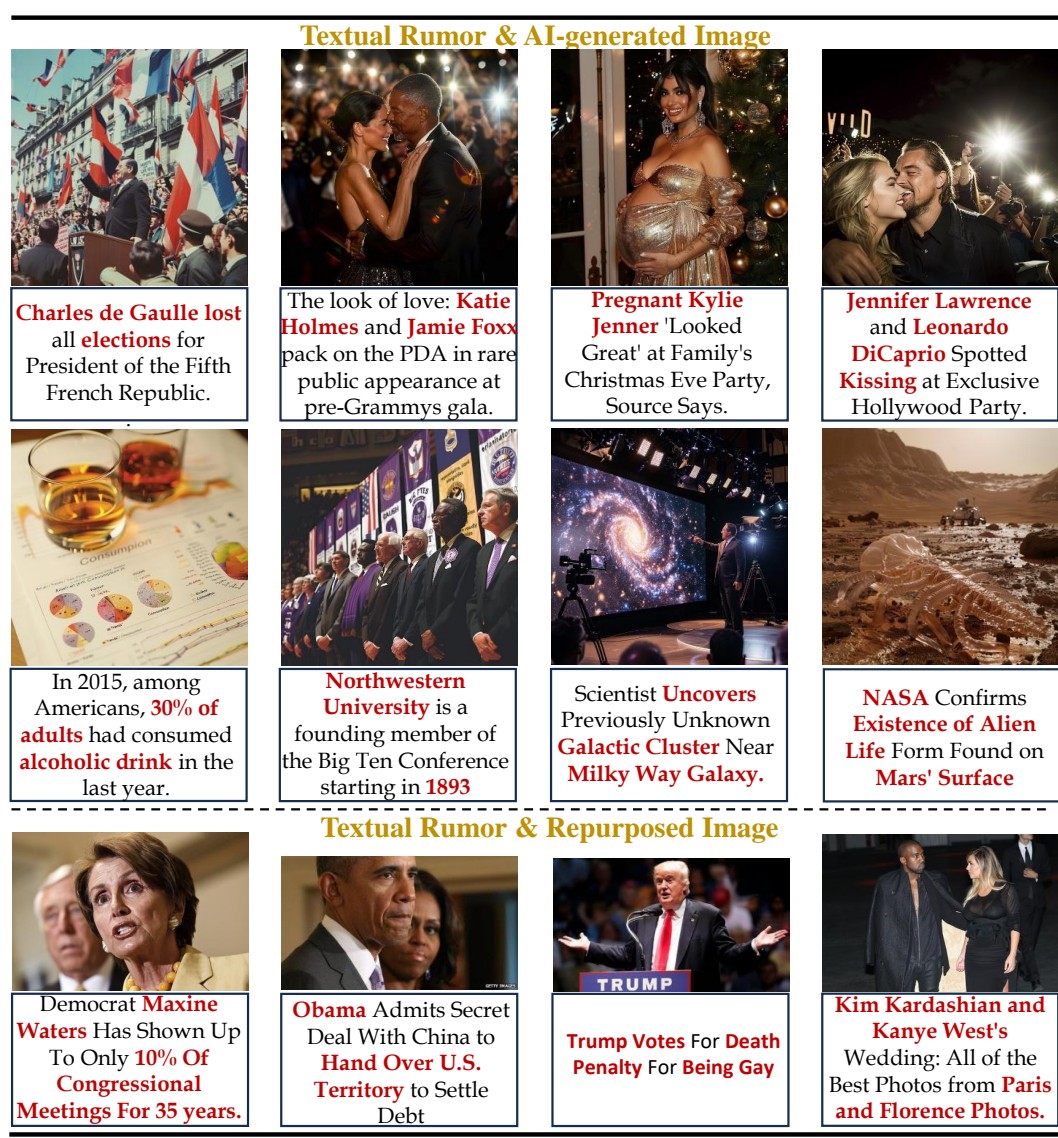

Figure 14: Visualization examples of multimodal misinformation from the textual veracity distortion.

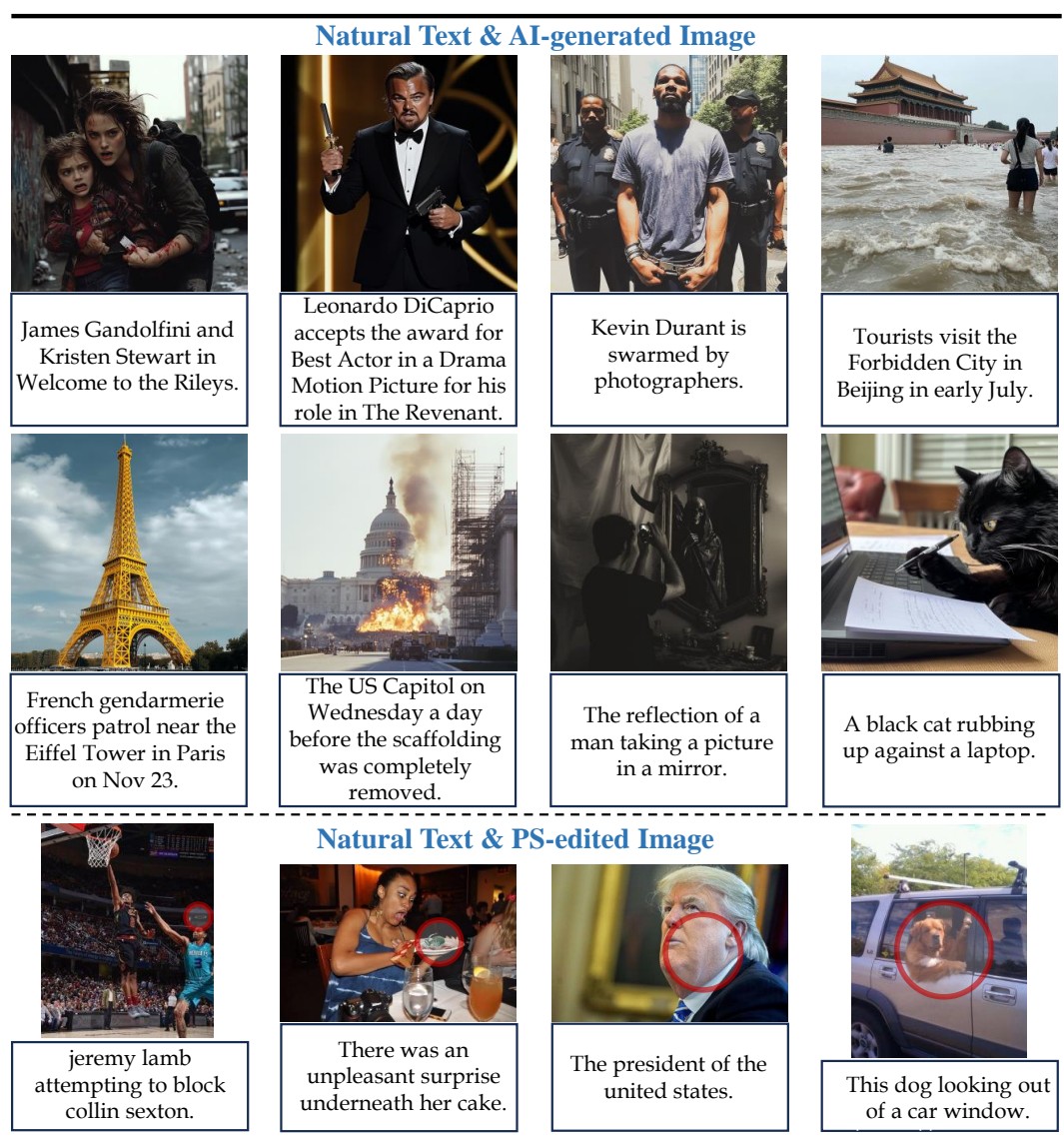

Figure 15: Visualization examples of multimodal misinformation from the visual veracity distortion.

## Repurposed Inconsistency

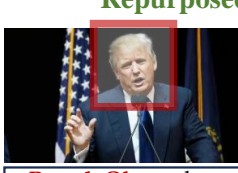

**Clinton** blames Bush administration for Senate job record

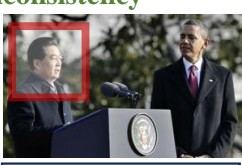

**Barack Obama** has called the Republican House Speaker urging resolution of a growing budget crisis

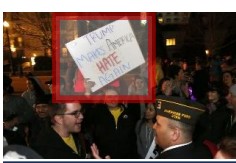

Obama and **Kenyatta** hold a joint news conference after their meeting

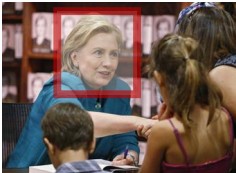

**Nationalists** also held a march in the western city of Lviv.

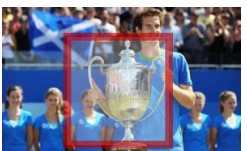

Secretary of State Hillary Rodham Clinton **adjusts her glasses** after answering a question

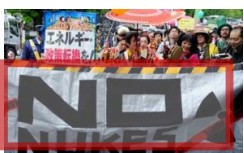

Andy Murray **returns the ball** to YenHsun Lu he won comfortably also in straight sets

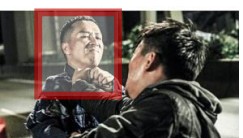

**MAPFRE fans show their support prior to the start of Leg 8**

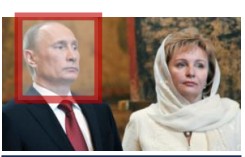

**The Raid 2 sees Iko Uwais** return as Rama a dogged and resourceful Indonesian cop

## Edited Inconsistency

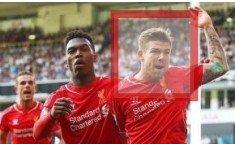

Vladimir Putin **celebrates** his wife Lyudmila in 2014

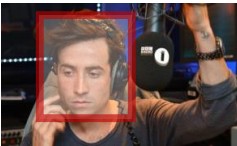

Liverpool s Alberto Moreno **cries** after scoring their third goal at Tottenham

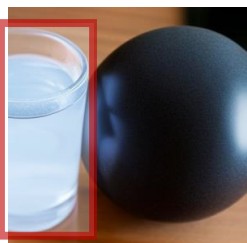

Jonathan Grado still **likes to work** six days a week tinkering with the various headphones Grado makes

A large black ball sitting next to a glass of **milk**.

Figure 16: Visualization examples of multimodal misinformation from the cross-modal consistency distortion.

