# OpenReview forum: "MMFakeBench: A Mixed-Source Multimodal Misinformation Detection Benchmark for LVLMs"
_ICLR.cc/2025/Conference — ICLR 2025 Poster_

### Official Review · Reviewer_RALP · 2024-10-27

**Soundness:** 3
**Presentation:** 4
**Contribution:** 3
**Rating:** 8
**Confidence:** 5

**Summary:**

This paper proposed MMFakeBench, a benchmark for multimodal disinformation detection from mixed sources. Unlike existing benchmarks focusing on single-source forgery, MMFakeBench approximates the real-world by encompassing three key sources: textual truthfulness distortions, visual truthfulness distortions, and cross-modal consistency distortions. complex disinformation. The benchmark contains 12 sub-categories of forgery types, totaling 3300 samples. The authors evaluate 6 current state-of-the-art detection methods and 15 LVLMs in a zero-shot setting, revealing the challenges faced by current methods in this mixed-source environment. To improve the detection accuracy, they propose MMD-Agent, which is based on the framework of LVLMs and divides the detection process into 2 phases, i.e., Hierarchical decomposition and Integration of internal and external knowledge.

**Strengths:**

1. Originality: The introduction of MMFakeBench addresses a previously unexplored field of multimodal false information detection, making it more representative of real-world scenarios.

2. Quality: The benchmark is carefully constructed, covering 12 types of forgery under three main categories, providing a robust evaluation platform.

3. Clarity: The structure of the paper is reasonable, and the categories included in the benchmark, the evaluation process, and the explanation of the proposed framework are clear and concise.

4. Significance: By revealing the limitations of current detection methods and proposing new frameworks, the paper has the potential to drive future research and improvement in this field.

**Weaknesses:**

1. Limited exploration of external knowledge sources: The paper acknowledges that reliance on sources such as Wikipedia is a limitation but can further explore its impact on detection performance and possible solutions.

2. Evaluation depth: Although the paper evaluated multiple models, it mainly focused on the zero-shot setting. If fine-tuning models or more ablation experiments are added, the experimental results may be more convincing.

3. The handling of rumors generated by GPT: The paper points out the challenges of detecting rumors generated by GPT, but further analysis or solutions can be proposed. There is no analysis of why GPT-generated Rumor is closer to Natural Rumor, or in other words, why is GPT-generated Rumor about as difficult to detect as Natural Rumor? After all, Artificial Rumor is also written by humans, so it should be about the same difficulty as Natural Rumor, but the experimental result is that Natural Rumor is the easiest to detect.

4. It is suggested to discuss and compare more related works such as [1] in this paper.

[1] Detecting and Grounding Multi-Modal Media Manipulation and Beyond. TPAMI 2024.

**Questions:**

1. External knowledge sources: Have you considered other external knowledge bases besides Wikipedia to enhance the detection of complex rumors?

2. Model tuning: Have you attempted to fine tune LVLMs on a subset of MMFakeBench to observe their performance improvement?

3. Practical application: How does MMD Agent perform in real-time detection scenarios? What is its computational requirement?

4. Scalability: Can your method be extended to other modalities, such as audio or video, to address more complex types of false information?

---

> ### Author Response · Authors · 2024-11-23
> **response to reviewer RALP (1)**
>
> Thank you for your thoughtful and constructive feedback. We are encouraged that you find the proposed benchmark addresses a previously unexplored field of multimodal misinformation detection, which is aligned with the core contribution in this work. Here is our response to address your concerns.
>
> > W1 and Q1 Limited exploration of external knowledge sources: explore its impact on detection performance and possible solutions. External knowledge sources: Have you considered other external knowledge bases besides Wikipedia to enhance the detection of complex rumors?*
>
> Thanks. We have incorporated additional experiments in **Table 3 (b) of the updated paper** and discussed analysis in **Section 5.3 (Ablation Study on Hierarchical Decomposition and Reasoning Knowledge) of the updated paper**. For convenience, we present the results in **Table r1**. Specifically,
>
> - Impact of external knowledge on detection performance. We conduct a comparative analysis of model performance with and without the integration of external knowledge sources (Wikipedia API). The detection performance (F1 score) of models without external knowledge exhibits a significant decrement in textual veracity distortion, decreasing from 37.6 to 18.0. This underscores **the critical role of external knowledge in validating textual veracity**.
>
> - Other external knowledge bases. To further evaluate the impact of alternative knowledge sources, we incorporated an additional resource: the Google Knowledge Graph API. The results also show that models **leveraging external knowledge outperform those relying solely on internal reasoning** for text checks, as the retrieved information provides essential context for assessing the veracity of claims.
>
> - Possible solutions. Developing robust, domain-specific external knowledge sources is essential. While general-purpose knowledge bases, such as those accessed via search engines like Wikipedia or Google, provide valuable context, their scope and depth are often limited when addressing the nuances of domain-specific misinformation. These limitations underscore the need for **curating more specialized knowledge repositories tailored to specific fields or applications**.
>
>
> Table r1: Ablation studies on the retrival external knowledge.
>
> |     | Real | TVD | VVD | CCD | Overall |
> | --- | --- | --- | --- | --- | --- |
> | With External Knowledge (Wikipedia) | 51.1 | **37.6** | 61.7 | 49.2 | 49.9 |
> | Without External Knowledge | 49.7 | **18.0** | 61.0 | 46.3 | 43.8 |
> | With External Knowledge (Google) | 50.2 | **33.4** | 62.2 | 48.1 | 48.5 |
>
> > W2 and Q2 Fine-tune LVLMs on a subset of MMFakeBench
>
> Thanks. We have included the experimental results of fine-tuning LVLMs on a subset of MMFakeBench in **Appendix Table 9 (b) of the updated paper**. For convenience, we present the results in **Table r2**. Specifically, we selected existing powerful specialized detectors, FKA-Owl [r1] which integrates LVLMs with two learnable networks for fine-tuning. FKA-Owl is initially trained on the DGM4 [r2] and we fine-tune it on the MMFakeBench validation set incrementally with 10, 100, and 1000 examples. We reported the results on both the DGM4 dataset and the MMFakeBench test set. The results indicate that:
>
> - **Tuning-based models can be hard to generalize to unseen forgery data.** For off-the-shelf dedicated detectors without fine-tuning, their performance on the proposed benchmark dataset is notably poor. With the rapid development of generative models, new forgery techniques and synthesized data continue to emerge. Models trained on limited samples face significant challenges in generalizing to unseen types of forgery data, highlighting a critical issue in the field of fake detection [r3], [r4], [r1].
>
> - **Catastrophic forgetting issues are inevitable.** Fine-tuning on new data improves performance on MMFakeBench but simultaneously leads to a decline in performance on the original dataset. This degradation underscores the phenomenon of catastrophic forgetting, where previously acquired knowledge, such as that from the DGM4 dataset, is progressively lost.
>
>
> Table r2: Performance (F1 score ↑) on different models when under few-shot fine-tuning.
>
> |     | MMFakeBench | DGM4 |
> | --- | --- | --- |
> | FKA-Owl before fine-tuning | 44.6 | 78.5 |
> | FKA-Owl after fine-tuning using 10 examples | 46.9 | 78.3 |
> | FKA-Owl after fine-tuning using 100 examples | 61.1 | 66.4 |
> | FKA-Owl after fine-tuning using 1000 examples | 76.2 | 64.8 |

---

> ### Author Response · Authors · 2024-11-23
> **response to reviewer RALP (2)**
>
> > W3 The handling of rumors generated by GPT: The paper points out the challenges of detecting rumors generated by GPT, but further analysis or solutions can be proposed. There is no analysis of why GPT-generated Rumor is closer to Natural Rumor, or in other words, why is GPT-generated Rumor about as difficult to detect as Natural Rumor? After all, Artificial Rumor is also written by humans, so it should be about the same difficulty as Natural Rumor, but the experimental result is that Natural Rumor is the easiest to detect.
>
> Thanks. We have expanded our analysis in **Section 5.3 (Analysis of Misinformation Sources and Types) of the updated paper**. The key analyses are summarized as follows:
>
> - **Rule-based nature of artificial rumors**. Artificial rumors are constructed by manually applying specific rules (such as sentence negation and key entity substitution) to modify Wikipedia sentences. Larger models like LLaVA-1.6-34B and GPT-4V perform relatively well on artificial rumors. This can be attributed to the fact that these rule-based patterns provide identifiable points of falsification, enabling verification through external knowledge sources, such as authoritative databases.
>
> - **Challenging nature of natural and GPT-generated rumors**. Natural rumors often originate from real-world news websites and social media while GPT-generated rumors contain human-like content with deceptive styles [r5] in domains, intentions and errors. These rumors leverage ambiguous language or vague claims that make detection particularly challenging, as they lack the structured inconsistencies seen in artificial rumors. Due to limited reasoning capabilities, open-source models even LLaVA-1.6-34B, are challenging to perform contextual reasoning to establish logical relationships and assess situational plausibility.
>
>
> > W4 It is suggested to discuss and compare more related works such as [1] in this paper.
>
> Thanks. We will include [Rui Shao, et al, Detecting and Grounding Multi-Modal Media Manipulation and Beyond] and any other relevant methods that may have been overlooked in the **Section 2 of the updated paper**. Specifically, recent research including [Detecting and Grounding Multi-Modal Media Manipulation and Beyond, TPAMI 2024], [SNIFFER: Multimodal Large Language Model for Explainable Out-of-Context Misinformation Detection. CVPR, 2024.], [FKA-Owl: Advancing Multimodal Fake News Detection through Knowledge-Augmented LVLMs. ACM MM 2024], has leveraged these pre-trained vision-language models which benefit from large-scale pre-training for reasoning context cues.
>
> > Q3 Practical application: How does MMD Agent perform in real-time detection scenarios? What is its computational requirement?
>
> Thanks. We would like to clarify that a discussion on the computational requirement of MMD-Agent has been included in **Appendix Section A.1.5 of the paper**. For convenience, the corresponding results are presented in **Table r4**. Specifically, we compare the inference time and computational resource usage of MMD-Agent with standard prompting methods. The detailed analysis are summarized as follows:
>
> - MMD-Agent introduces higher inference time compared to standard prompt methods, primarily due to the additional reasoning steps, such as extracting key entities from the text.
>
> - MMD-Agent does not lead to additional GPU memory consumption. This is because the Agent method does not affect the model parameter deployment and dataset storage.
>
>
> While the increased inference time is a consideration, **the substantial gains brought by the MMD-Agent should not be overlooked**:
>
> - **Significant performance gain.** The MMD-Agent method greatly enhances the multimodal misinformation detection performance, with F1 score improving from 25.7 to 49.9.
>
> - **Excellent interpretability.** The MMD-Agent method offers stronger interpretability since MMD-Agent offers a detailed analysis of the misinformation rather than only providing classification labels as used in standard prompt methods. Specifically, as shown in Fig.3 of the main paper, the content of the intermediate rationales such as Thought 2, Thought 3, and Thought 4 accurately analyzes the specific reasons for misinformation from different sources.
>
>
> Table r4: Computational requirement of different methods on LLaVA-1.6-34B.
>
> |     | Standard Prompting | MMD-Agent |
> | --- | --- | --- |
> | Average Inference Time (s) ↓ | 5.97s | 49.04s |
> | Memory (GB) ↓ | 82G | 82G |
> | Performance (F1 score) ↑ | 25.7 | 49.9 |

---

> ### Author Response · Authors · 2024-11-23
> **response to reviewer RALP (3)**
>
> > Q4 Can your method be extended to other modalities, such as audio or video, to address more complex types of false information?
>
> Thanks. Our current benchmark focuses on image and text modalities. However, as you insightly suggested, audio and video play crucial roles in real-world misinformation dissemination. **We plan to explore incorporating audio and video data samples in future work to expand our benchmark**, making the task more reflective of real-world scenarios. Additionally, with the advancement of multimodal generative models such as ChatGPT and EMU3 [r6], these models have demonstrated the ability to simultaneously process multiple data modalities. Our method can **make it well-suited for integration with emerging multimodal generative models** to address complex misinformation scenarios that involve multimodal data.
>
> [r1] Liu X, et al. FKA-Owl: Advancing Multimodal Fake News Detection through Knowledge-Augmented LVLMs. ACM MM, 2024.
>
> [r2] Shao R, et al. Detecting and Grounding Multi-Modal Media Manipulation. CVPR, 2023.
>
> [r3] Akhtar M, et al. MDFEND: Multi-domain Fake News Detection. CIKM, 2021.
>
> [r4] Ojha U, et al. Towards Universal Fake Image Detectors That Generalize Across Generative Models. CVPR, 2023.
>
> [r5] Chen C, et al. Can LLM-Generated Misinformation Be Detected?. ICLR, 2024.
>
> [r6] Wang X, et al. Emu3: Next-Token Prediction is All You Need. 2024.

---

> ### Comment · Reviewer_RALP · 2024-11-26
>
> Thanks for your response. Most of my concerns are solved. I recommend the acceptance.

---

> > ### Author Response · Authors · 2024-11-26
> > **Response to Official Comment by Reviewer RALP**
> >
> > Thanks for your comment! We are glad that our answer addresses your question! Based on your suggestions, we have revised our paper to further enhance its clarity and quality.

---

### Official Review · Reviewer_x4j6 · 2024-11-01

**Soundness:** 3
**Presentation:** 3
**Contribution:** 3
**Rating:** 5
**Confidence:** 4

**Summary:**

This paper proposed a new benchmark of multimodal misinformation detection, and the authors constructed a multimodal misinformation data generation pipeline by integrating characteristics from previous multimodal misinformation benchmarks. Additionally, the authors proposed a novel framework named MMd-Agent, which decomposes the multimodal misinformation detection task into three sub-tasks based on the three critical sources (textual veracity distortion,visual veracity distortion, and cross-modal consistency distortion) identified in this paper and processes them in a step-by-step manner, this method finally achieved good results.

**Strengths:**

1.	From a mixed-source perspective, this paper proposes a new benchmark for multimodal misinformation detection, addressing some weaknesses in existing datasets.
2.	The authors proposed a novel framework named MMd-Agent, which achieved impressive performance.
3.	The paper is written in an easily understandable manner, and the experimental evaluations are thorough.

**Weaknesses:**

1.	This paper did not present the existing high-impact multimodal misinformation detection benchmarks or compare them to the proposed benchmark concerning the three critical sources.. Could the authors list some relevant multimodal misinformation benchmarks, multimodal misinformation benchmarks detection methods, and SOTA？
2.	This paper did not present a summary of mainstream methods for detecting multimodal misinformation and summarize the classifications of these methods？
3.	The experimental analysis lacks depth; for example, it only addresses some performance limitations of LLaVA-1.6-34B and GPT (as mentioned in the Error Analysis), while other models are evaluated solely on their performance. I hope the authors can conduct a more thorough analysis of the experimental section and consider analyzing the shortcomings of larger models as well.
4.	The novelty of the proposed MMd-Agent is somewhat limited.

**Questions:**

1.	What’s the distinction between Natural Rumor and Artificial Rumor?
2.	Do the definitions of the three critical sources apply to other publicly available datasets?
3.	The generation style of the visual veracity distortion data is relatively uniform. Does this meet the benchmark's intended purpose?
4.	Whether the accuracy of all GPT-generated rumors has been manually verified is necessary.
5.	Has the impact of dataset imbalance (between real and fake, as well as imbalances among different subclasses) been validated? This makes it difficult to reuse the dataset.
6.	The authors should provide further analysis of the experimental results in Figure 5, comparing their performance differences to better illustrate the distinctions between different categories(Natural Rumor, Artificial Rumor and GPT-generated Rumor).
7.	I believe that mixing human-generated and LLM-generated misinformation is very meaningful; however, other datasets, such as Twitter and GossipCop, also contain three critical categories. How are the limitations of the different datasets in Table 1 assessed?

---

> ### Author Response · Authors · 2024-11-23
> **response to reviewer x4j6 (1)**
>
> We want to express our great thanks for your valuable suggestions and patience regarding our work. We have carefully addressed each of your comments and provided a point-by-point response below.
>
> > *Q1 This paper did not present the existing high-impact multimodal misinformation detection benchmarks or compare them to the proposed benchmark concerning the three critical sources.. Could the authors list some relevant multimodal misinformation benchmarks, multimodal misinformation benchmarks detection methods, and SOTA？*
>
> Thanks. We have compared existing misinformation datasets with proposed benchmarks in **Table 1 of the main paper**. Additionally, we have listed some detection methods and their performance concerning the three critical sources.
>
> **Textual Veracity Distortion**: We reference **Politifact** [r1] and **Gossipcop** [r1], two widely used datasets that primarily evaluate veracity of textual claims. We also list three corresponding methods UPFD [r2], DECOR [r3] and current SOTA CPNM [r4].
>
> Table r1: Performance (F1 score ↑) of different models on textual veracity distortion dataset including Politifact and Gossipcop.
>
> |     | UPFD | DECOR | CPNM |
> | --- | --- | --- | --- |
> | Politifact | 84.65 | 94.79 | 98.10 |
> | Gossipcop | 97.22 | 90.31 | 97.30 |
>
> **Visual Veracity Distortion:** Existing methods in Synthetic Image Detection typically detect uni-modal images generated by various diffusinon models such as LDM [r5]. We also list three corresponding methods UnivFD [r6], RINE [r7] and current SOTA FatFormer [r8].
>
> Table r2: Performance (ACC score ↑) of different models on images generated by latent diffusion models (LDM).
>
> |     | UnivFD | RINE | FatFormer |
> | --- | --- | --- | --- |
> | LDM | 87.4 | 95.0 | 97.4 |
>
> **Cross-modal Consistency Distortion:** For multimodal misinformation that spans across modalities, existing methods focus on capturing cross-modal inconsistency. We consider the DGM4 dataset [r9], which manipulates on both images and text. Additionally, we list three corresponding methods HAMMER [r9], UFAormer [r10] and current SOTA EMSF [r11].
>
> Table r3: Performance (AUC score ↑) of different models on HAMMER dataset.
>
> |     | HAMMER | UFAFormer | EMSF |
> | --- | --- | --- | --- |
> | DGM4 | 93.19 | 93.81 | 95.11 |
>
> Based on the relevant benchmarks mentioned above, the key distinctions are summarized as follows:
>
> - **Mixed-source multimodal misinformation.** Existing datasets primarily focus on single-source misinformation including text-based rumors (e.g., Politifact [r1] and Gossipcop [r1]) or cross-modal inconsistencies (e.g., DGM4 [r9]). While useful for addressing specific tasks, these datasets can not handle the mixed-source scenarios, which are prevalent in real-world settings. To bridge this gap, our proposed benchmark integrates three primary forgery categories: textual veracity distortion, visual veracity distortion, and cross-modal consistency distortion, encompassing 12 subcategories of forgery types.
>
> - **Integration of text-image rumors.** Textual veracity distortion datasets (e.g., Politifact [r1] and Gossipcop [r1]), typically focus exclusively on text-based rumors. In contrast, our dataset incorporates text-image rumors using highly relevant real or AI-generated images to simulate real-world scenarios.
>
> - **Advanced generative models.** Current datasets often overlook the evolving threat posed by advanced generative models. Our dataset construction actively employs a diverse range of generative models and AI tools to create sophisticated forgeries.
>
>
> We would like to inquire whether the dataset and detection methods we have provided align with your specific requirements. If not, we would greatly appreciate any concrete examples you could share to better illustrate your needs.

---

> ### Author Response · Authors · 2024-11-23
> **response to reviewer x4j6 (2)**
>
> > *Q2 This paper did not present a summary of mainstream methods for detecting multimodal misinformation and summarize the classifications of these methods？*
>
> Thanks. We have presented a summary of mainstream methods for detecting multimodal misinformation in **Section 2 of the updated paper**. Multimodal misinformation detection methods adhere to fuse cross-modal features to extract semantic representations, enabling robust identification of inconsistencies across modalities. Based on their model frameworks, these methods can be divided into two categories:
>
> - **Small-scale Attention-based Networks.** Early research primarily focus on designing attention-based networks [r12], [r13], [r14] with diverse learning strategies [r15], [r16] to effectively capture cross-modal interactions. For instance, Coattention network [r12], contextual attention network [r13] and improved Multi-gate Mixture-of-Expert networks (iMMoE) [r14] are proposed to better refine and fuse textual and visual features. Ambiguity learning [r15] and causal reasoning [r16] are separately introduced to address the issue of modal disagreement decisions and spurious correlation in data bias.
>
> - **Vision-Language Pretraining as Foundation Models.** With the advancements of large-scale vision-language pretraining (e.g., CLIP, ALIGN, and recent LVLMs), recent research [r9], [17], [18] has shifted towards leveraging these pre-trained architectures as foundational models for multimodal misinformation detection. These methods benefit from the broad world knowledge and robust cross-modal reasoning learned during pretraining.
>
>
> > *Q3 The experimental analysis lacks depth; for example, it only addresses some performance limitations of LLaVA-1.6-34B and GPT (as mentioned in the Error Analysis), while other models are evaluated solely on their performance. I hope the authors can conduct a more thorough analysis of the experimental section and consider analyzing the shortcomings of larger models as well.*
>
> Thanks. We have **included more error analyses in Section A.1.7 of the updated Appendix**. These case studies span four models of different scales (i.e., GPT-4V, LLaVA-1.6-34B, BLIP2-FLAN-T5-XXL, and VILA-13B) and on three distinct forgery sources of textual veracity distortions, visual veracity distortion and cross-modal consistency distortion. Based on these cases, we provide a deep analysis of the shortcomings of both largely and moderately sized models when encountering different sources of multimodal misinformation. Specifically,
>
> (1) textual veracity distortion:
>
> - **Reliance on external knowledge.** While Largly sized models like LLaVA-1.6-34B and GPT-4V exhibit strong reasoning capabilities, **they are challenging to infer the factual correctness of a statement without access to a broader context**. For instance, when faced with factually incorrect statements that appear linguistically accurate, such as the GPT-generated rumor, these four models may lack the capability to question the information without retrieving corroborative evidence.
>
> (2) Visual Veracity Distortion.
>
> - **Limited Sensitivity to Abnormal Physical Features**: Models including GPT-4V, LLaVA-1.6-34B, BLIP2-FLAN-T5-XXL, and VILA-13B, **face challenges in discerning abnormal physical characteristics**. For instance, in the PS-edited examples, these four models fail to detect subtle yet unrealistic manipulations, such as swollen necks or distorted facial features in images of Donald Trump. Instead, they are frequently distracted by more prominent visual elements, such as facial expressions or gestures, resulting in misjudgments when identifying these manipulations.
>
> (3) Cross-modal Consistency Distortion.
>
> - **Distraction by global consistent semantics.** When confronted with scenarios where images and text exhibit only subtle inconsistencies while other aspects remain largely consistent, large-scale models such as LLaVA-1.6-34B and GPT-4V often struggle to detect these discrepancies. **These models can be distracted by the dominant presence of consistent content**, which obscures the subtle mismatches critical for accurate misinformation detection. For instance, in the case of repurposed inconsistency, all four models focus on global semantics, such as a person delivering a speech, which divert attention from deeper, subtle inconsistencies in the scenario.

---

> ### Author Response · Authors · 2024-11-23
> **response to reviewer x4j6 (3)**
>
> > *Q4 The novelty of the proposed MMd-Agent is somewhat limited.*
>
> Thanks. We highlight the novelty of the proposed MMD-Agent as follows:
>
> - **Unique design for addressing mixed-source multimodal misinformation**. Current LVLMs lack the task-specific precision required for addressing mixed-source multimodal misinformation. To tackle this, we present MMD-Agent, a modular framework that **decomposes detection into three smaller subtasks**: textual, visual, and cross-modal checks. **Each subtask is addressed through an interleaved sequence of reasoning and action, and integrating external knowledge sources like Wikipedia,** ensuring accurate and reliable detection. These innovations establish MMD-Agent as a benchmark solution for the complexities of mixed-source misinformation on MMFakeBench. As a result, MMD-Agent significantly improves the F1-score for all selected models. Notably, **LLaVA-1.6-34B using MMD-Agent achieves an F1-score of 49.9%, approaching the 51% score of GPT-4V with the standard prompting**.
>
> - **Interpretability.** MMD-Agent **offers stronger interpretability since it offers a detailed analysis** of the misinformation rather than only providing classification labels as used in standard prompting. Specifically, as shown in Fig.4 of the main paper, the content of the intermediate rationales such as Thought 2, Thought 3 and Thought 4 accurately analyzes the specific reasons for misinformation from different sources.
>
> - **Scability.** The scalable MMD-Agent allows the framework to **add domain-specific reasoning components or task-specific acting modules** without significantly increasing system complexity. This scalability enables the system to **reason over dynamically updated knowledge bases**, ensuring adaptability to evolving information landscapes. Additionally, MMD-Agent **operates independently of model fine-tuning**, making it well-suited for integration with emerging multimodal generative models.
>
>
> > *Q5 What’s the distinction between Natural Rumor and Artificial Rumor?*
>
> Thanks. We address the distinction between natural rumor and artificial rumor as follows:
>
> - Artificial Rumors are intentionally constructed by **manually applying specific rules to modify statements**, thereby creating deviations from objective truth. In our benchmark, this type of data is generated by modifying Wikipedia sentences using techniques such as sentence negation, key entity substitution, and word replacement etc.
>
> - Natural Rumors, such as those commonly found on news websites and social media platforms, are inherently challenging to detect **due to their lack of strict rules or predefined patterns**. These rumors frequently **draw on current events or trending topics, giving them an appearance of relevance and authenticity**. Our benchmark includes fabricated political rumors or entertainment news that have been widely shared online.
>
>
> All of them serve as valuable test cases for evaluating the robustness and generalizability of misinformation detection systems.
>
> > *Q6 The generation style of the visual veracity distortion data is relatively uniform. Does this meet the benchmark's intended purpose?*
>
> Thanks. Our dataset is designed with a high degree of diversity, ensuring it to meet the benchmark's intended purpose. To better illustrate style diversity, we have incorporated **more visualized examples of visual veracity distortions in Appendix Fig. 15 of the updated paper**. Specifically, Visual veracity distortion in our benchmark incorporates two primary techniques: PS-edited and AI-generated content. These techniques are deliberately guided by diverse themes, intentions, or scenarios, ensuring that the generation styles are far from uniform：
>
> - PS-edited manipulations involve the deliberate alteration of visual content to mislead or misrepresent reality by targeting specific elements within an image. These manipulations can categorized into **person-based adjustments** and **object-based modifications**. Person-based adjustments include changes to physical attributes, such as altering facial features, expressions, or postures. Object-based modifications, involve inserting, removing, or altering elements within the scene to misrepresent its content.
>
> - AI-generated images are meticulously crafted by creating **fabricated scenarios with well-known entities** or **counterfactual elements**. Well-known entities including celebrities and landmarks, are depicted in ways that contradict social norms (e.g., violence, arrest), attributes, and events. Counterfactual elements, meanwhile, are incorporated to reinforce these fabrications, creating a false sense of plausibility or dramatic impact by embedding elements that conflict with the real-world context.

---

> ### Author Response · Authors · 2024-11-23
> **response to reviewer x4j6 (4)**
>
> > *Q7 Whether the accuracy of all GPT-generated rumors has been manually verified is necessary.*
>
> Thanks. We would like to clarify that GPT-generated rumors included in our dataset have undergone manual verification to ensure their accuracy. The verification process was designed with two primary objectives:
>
> - **Ensuring the presence of false statements.** To guarantee that the generated rumors indeed contain statements that contradict objective facts, we employ 10 experienced annotators which can utilize online tools such as web searches and fact-checking platforms. For each rumor, the annotators cast their votes on its veracity, and only examples with a majority consensus indicating false information is included in the final dataset.
>
> - **Reducting social harm.** The annotators are also tasked with meticulously reviewing the generated rumors to identify and exclude high-risk texts, such as those involving sensitive topics such as politics and race. This process was essential to ensure the safety, ethical integrity, and responsible use of the final dataset.
>
>
> > *Q8 Has the impact of dataset imbalance (between real and fake, as well as imbalances among different subclasses) been validated? This makes it difficult to reuse the dataset.*
>
> Thanks. Since our benchmark is designed to support a multi-class evaluation, we ensured a balanced distribution of data across the different forgery sources and forgery sub-methods, as illustrated in **Fig. 2 of the main paper**. Specifically, we **proportionally distribute the data across the four sources based on the number of forgery subtypes**. Specifically, the dataset comprises 30% real data, 30% textual veracity distortion with five forgery subtypes, 10% visual veracity distortion with two forgery subtypes, and 30% cross-modal consistency distortion with five forgery subtypes.
>
> Specifically, the dataset composition is as follows:
>
> - **Balance between six data sources to constitute real data:** The real dataset is constructed by equally sampling from six corresponding data sources, including The Guardian, BBC, USA TODAY, The Washington Post, MS-COCO, and Fakeddit.
>
> - **Balance between three types of textual rumors to constitute textual veracity distortion:** This category comprises three key types of textual rumors: natural rumor, artificial rumor and GPT-generated rumor, paired with two types of supporting images. To ensure balance, equal proportions of samples are allocated among the three textual rumor types.
>
> - **Balance between two visual forgery subtypes to constitute visual veracity distortion**: This category includes two types of forgery subtypes: PS-edited and AI-generated images. The dataset was evenly split between these two types to maintain a balanced representation of visual distortions.
>
> - **Balance between two types of cross-modal inconsistency to constitute cross-modal consistency distortion**: This category encompasses two primary inconsistency types: repurposed inconsistency and edited inconsistency. To maintain balance, the dataset is evenly distributed between repurposed and edited inconsistencies. Additionally, repurposed inconsistency encompasses three query-based subtypes in balance.
>
>
> > *Q9 The authors should provide further analysis of the experimental results in Figure 5, comparing their performance differences to better illustrate the distinctions between different categories(Natural Rumor, Artificial Rumor and GPT-generated Rumor).*
>
> Thanks. **We have expanded our analysis in Section 5.3 (Analysis of Misinformation Sources and Types) of the updated paper**. The key analyses are summarized as follows:
>
> - **Rule-based nature of artificial rumors**. Artificial rumors are constructed by manually applying specific rules (such as sentence negation and key entity substitution) to modify Wikipedia sentences. Larger models like LLaVA-1.6-34B and GPT-4V perform relatively well on artificial rumors. This can be attributed to the fact that these rule-based patterns provide identifiable points of falsification, enabling verification through external knowledge sources, such as authoritative databases.
>
> - **Challenging nature of natural and GPT-generated rumors**. Natural rumors often originate from real-world news websites and social media while GPT-Generated Rumors contain deceptive styles [r19] in domains, intentions and errors. These rumors leverage ambiguous language or vague claims that make detection particularly challenging, as they lack the structured inconsistencies seen in artificial rumors. Due to limited reasoning capabilities, open-source models even LLaVA-1.6-34B, are challenging to perform contextual reasoning to establish logical relationships and assess situational plausibility.

---

> ### Author Response · Authors · 2024-11-23
> **response to reviewer x4j6 (5)**
>
> > *Q10 I believe that mixing human-generated and LLM-generated misinformation is very meaningful; however, other datasets, such as Twitter and GossipCop, also contain three critical categories. How are the limitations of the different datasets in Table 1 assessed?*
>
> Thanks. We would like to clarify that **Table 1 in our paper** explicitly annotates the number of rumor types present in different datasets. Existing datasets are limited by their focus on a single type of textual rumor. In Table r6, We also show the specific types of rumors in the relevant datasets. Specifically, the **FEVER dataset exclusively contains artificial rumors** generated through manual modification of Wikipedia sentences. The **Politifact and Gossipcop datasets include only natural rumors**, derived from political and gossip news. The **LLMFake dataset is specifically designed to address misinformation generated by LLMs**.
>
> In contrast, our proposed dataset **encompasses three distinct types of textual veracity distortions**—artificial, natural and LLM-generated misinformation—providing a more comprehensive framework for evaluation and analysis.
>
> Table r6: Rumor types are incorporated in existing datasets.
>
> | Dataset | Artificial Rumor | Natural Rumor | GPT-generated Rumor |
> | --- | --- | --- | --- |
> | FEVER | √   | ×   | ×   |
> | Politifact | ×   | √   | ×   |
> | Gossipcop | ×   | √   | ×   |
> | Snopes | ×   | √   | ×   |
> | MOEHEG | ×   | √   | ×   |
> | LLMFake | ×   | ×   | √   |
> | MMFakeBench | √   | √   | √   |
>
> [r1] Shu K, et al. FakeNewsNet: A Data Repository with News Content, Social Context, and Spatiotemporal Information for Studying Fake News on Social Media. Big Data, 2020.
>
> [r2] Dou Y, et al. User Preference-aware Fake News Detection. SIGIR, 2021.
>
> [r3] Wu J, et al. DECOR: Degree-Corrected Social Graph Refinement for Fake News Detection. KDD, 2023.
>
> [r4] Donabauer G, et al. Challenges in Pre-Training Graph Neural Networks for Context-Based Fake News Detection: An Evaluation of Current Strategies and Resource Limitations. COLING, 2024.
>
> [r5] Rombach R, et al. High-Resolution Image Synthesis With Latent Diffusion Models. CVPR, 2022.
>
> [r6] Ojha U, et al. Towards Universal Fake Image Detectors That Generalize Across Generative Models. CVPR, 2023.
>
> [r7] Koutlis C, et al. Leveraging Representations from Intermediate Encoder-Blocks for Synthetic Image Detection. ECCV, 2024.
>
> [r8] Liu H, et al. Forgery-aware Adaptive Transformer for Generalizable Synthetic Image Detection. CVPR, 2024.
>
> [r9] Shao R, et al. Detecting and Grounding Multi-Modal Media Manipulation. CVPR, 2023.
>
> [r10] R, et al. Unified Frequency-Assisted Transformer Framework for Detecting and Grounding Multi-modal Manipulation. IJCV, 2024.
>
> [r11] Wang J, et al. Exploiting Modality-Specific Features for Multi-Modal Manipulation Detection and Grounding. ICASSP, 2024.
>
> [r12] Wu Y, et al. Multimodal fusion with co-attention networks for fake news detection. ACL Findings, 2021.
>
> [r13] Qian S, et al. Hierarchical multi-modal contextual attention network for fake news
>
> detection. SIGIR, 2021.
>
> [r14] Qian S, et al. Bootstrapping Multi-view Representations for Fake News Detection. AAAI, 2023.
>
> [r15] Chen Y, et al. Cross-modal ambiguity learning for multimodal fake news detection. WWW, 2022.
>
> [r16] Chen Z, et al. Causal intervention and counterfactual reasoning for multi-modal fake news detection. ACL, 2023.
>
> [r17] Liu, X, et al. FKA-Owl: Advancing Multimodal Fake News Detection through Knowledge-Augmented LVLMs. ACM MM, 2024.
>
> [r18] Qi P, et al. SNIFFER: Multimodal Large Language Model for Explainable Out-of-Context Misinformation Detection. CVPR, 2024.
>
> [r19] Chen C, et al. Can LLM-Generated Misinformation Be Detected?. ICLR, 2024.

---

> ### Author Response · Authors · 2024-11-25
> **Looking forward to the response from Reviewer x4j6**
>
> Dear Reviewer x4j6,
>
> We have tried our best to address all the concerns and provided as much evidence as possible. May we know if our rebuttals answer all your questions? We truly appreciate it.
>
> Best regards,
>
> Author #6524

---

> > ### Comment · Reviewer_x4j6 · 2024-12-01
> > **Keep my rating**
> >
> > Thanks for your detailed response, which addresses partial concerns. I still keep my original rating. Thanks.

---

### Official Review · Reviewer_onkv · 2024-11-02

**Soundness:** 2
**Presentation:** 3
**Contribution:** 2
**Rating:** 6
**Confidence:** 4

**Summary:**

The authors introduce MMFakeBench, presented as a benchmark for evaluating mixed-source multimodal misinformation detection (MMD). They develop a multi-class evaluation metric and conduct evaluations of six state-of-the-art detection methods and 15 large vision-language models on MMFakeBench. Additionally, the authors propose an LVLM-based framework, MMD-Agent, which is intended to enhance detection performance in mixed-source MMD tasks.

**Strengths:**

The paper offers relatively comprehensive coverage of misinformation types, utilizes a diverse set of detection methods, and includes clear figures explaining the dataset and workflow.

**Weaknesses:**

1. The contributions of this work are relative limited. Compared to prior benchmarks like DGM4, the primary difference lies in the use of large models to generate data, without introducing novel, highly deceptive misinformation generation methods. Additionally, the proposed MMD-Agent framework contributes minimally to the field, relying heavily on the use of LVLMs.

2. The dataset size is relatively small. While recent benchmarks like DGM4 contain over 200k samples, MMFakeBench includes only about 10k. This raises concerns about whether each misinformation type is represented with enough diversity to support model generalization.

3. It is unclear whether the misinformation types included are meaningful. For example, the Cross-modal Consistency Distortion example in Figure 2 — “An old man reading a book on a park bench” versus “An old man reading a newspaper on a park bench” — lacks practical relevance. Emphasis should be placed on edits involving named entities, such as celebrities, to ensure sufficient impact and deception. Similarly, the Visual Veracity Distortion example with “Veracious Text & AI-generated Image” lacks sufficient misleading potential; the reader’s first impression might be that the image is obviously edited, which fails to show it can lead viewers into perceiving falsified content as authentic.

**Questions:**

1. For detecting outputs from generative models, a CNN often achieves better detection performance than an LVLM. Consider including performance results using foundation models paired with a fine-tuned network.

2. Some details in Table 1 are unclear. For instance, why do MEIR and DGM4, which involve text editing, lack Textual Veracity Distortion? Edited texts are essentially a type of rumour.

Based on the clarification regarding the proposed dataset as an evaluation resource and the additional explanations provided, I believe you have addressed most of my concerns and I have decided to raise my rating to 6. However, as MMFakeBench is mix-sourced, compared to NewsCLIPpings, which focuses on OOC detection, more data is needed to achieve sufficient diversity and ensure the validity of experimental results. I look forward to your future work.

---

> ### Author Response · Authors · 2024-11-23
> **response to reviewer onkv (1)**
>
> We want to express our great thanks for your feedback and patience regarding our work. We have carefully addressed each of your comments and provided a point-by-point response below.
>
> > *Q1 The contributions of this work are relative limited. Compared to prior benchmarks like DGM4, the primary difference lies in the use of large models to generate data, without introducing novel, highly deceptive misinformation generation methods. Additionally, the proposed MMD-Agent framework contributes minimally to the field, relying heavily on the use of LVLMs.*
>
> Thanks. The key contribution of this work lies in addressing the **real-world critical yet underexplored issue** of the con-existence of multiple forgery sources, and proposes **the first benchmark explicitly designed for evaluating mixed-source multimodal misinformation**. We compare our benchmark with the prior dataset like DGM4 to further highlight the contributions of this work:
>
> - **Mixed forgery sources**. Existing datasets typically assume that the forgery source is predefined and confined to a single source, such as text-only rumors, image-only fabrications or image-text inconsistencies (e.g., DGM4) , as identified by **Table 1 of the main paper**. **These assumptions overlook the real-world scenarios, where the multiple forgery source is often co-exist**.
>
> - **Comprehensive forgery types.** The DGM4 dataset implements edit-based manipulation to alter the cross-modal inconsistencies. The proposed dataset encompasses 12 subcategories of forgery types, covering methods including **text/image editing**, **text/image repurposing**, **text/image generation using LLMs or Diffusion models**, as well as forgery samples collected from real-world platforms.
>
> - **Collaborative generation with large-scale generative models.** Small-scale models in the DGM4 dataset are often applied to specific tasks, limiting style diversity in forgery data. In contrast, large-scale generative models have revolutionized data generation with high fidelity and diversity. To increase controllability in producing application-specific misinformation datasets, we propose an **innovative data generation pipeline** that integrates state-of-the-art generative models and AI tools. This pipeline systematically generates diverse misinformation data, including **textual rumors**, **enriched contextual descriptions**, **fact-conflicting elements**, and **high-fidelity images**, aligned with various **intents, topics, and real-world scenarios**. It is noted that the use of large-scale generative models for data generation is a cutting-edge research area[r2], [r3]. The data generation method we propose offers valuable inspiration and practical utility for similar domains, producing valuable data that advances multimodal misinformation detection tasks.
>
>
> **The Advancement of MMD-Agent.** With broad pretraining and strong reasoning capabilities, LVLMs demonstrate notable generalizability in misinformation detection [r3], [r4], [r5]. However, their application often **lacks the task-specific precision** required for addressing mixed-source multimodal misinformation. To tackle this, we present MMD-Agent, a modular framework that **decomposes detection into three smaller subtasks** to textual, visual, and cross-modal checks. Each subtask is **addressed through an interleaved sequence of reasoning and action**, and integrating external knowledge sources like Wikipedia, ensuring accurate and reliable detection. MMD-Agent is the **first approach to leverage Agent-based methods** for addressing multimodal misinformation detection. As a result, MMD-Agent significantly improves the F1-score for all selected models. Notably, **LLaVA-1.6-34B using MMD-Agent achieves an F1-score of 49.9%, approaching the 51% score of GPT-4V** with the standard prompting. These innovations establish MMD-Agent as a benchmark solution on MMFakeBench.
>
> In summary, our benchmark incorporates comprehensive and realistic misinformation scenarios, marking a significant step toward addressing misinformation in practical applications.

---

> ### Author Response · Authors · 2024-11-23
> **response to reviewer onkv (2)**
>
> > *Q2 The dataset size is relatively small. While recent benchmarks like DGM4 contain over 200k samples, MMFakeBench includes only about 10k. This raises concerns about whether each misinformation type is represented with enough diversity to support model generalization.*
>
> Thanks. Unlike DGM4 [r1], which is commonly used as a training dataset, **MMFakeBench is explicitly designed as an evaluation dataset**, a distinction also recognized by Reviewer jzRM.
>
> - **Comparable in scale to other LVLM-based evaluation datasets**. Notably, there are numerous carefully designed evaluation datasets specifically developed to benchmark the capabilities of LVLMs across various tasks. In Table r1, we compare the scales of different evaluation datasets, including test sets from two multimodal misinformation datasets (DGM4 [r1] and NewsCLIPpings [r6]) and three representative LVLM-based evaluation datasets (MMMU [r7], MMSatetyBench [r8], OCRBench [r9]). As an evaluation dataset, **MMFakeBench is comparable in scale to other evaluation datasets and is well-suited for assessing model performance**.
>
> - **Enough diversity to evaluate model generalization.** MMFakeBench comprises **12 distinct forgery types** across textual, visual, and cross-modal forgery sources. Each forgery type is carefully curated to represent **a wide range of real-world misinformation scenarios**. Textual Veracity Distortion: Includes natural rumors **sourced from fact-checking platforms**, artificially generated rumors through **manual edits**, and GPT-generated rumors created **using diverse prompting strategies across multiple domains**. Visual Veracity Distortion: Combines high-quality AI-generated images (e.g., MidJourney) with rigorously selected Photoshop-edited images. Cross-Modal Consistency Distortion: Features mismatched text-image pairs derived from **repurposed text/images** or generated via **InstructPix2Pix**, simulating subtle and complex inconsistencies to test advanced reasoning capabilities.
>
>
> Table r1: Evaluation data size on different datasets.
>
> |     | DGM4 (Training) | NewsCLIPpings (Training) | MMMU (Evaluation) | MMSafetyBench (Evaluation) | OCRBench (Evaluation) | MMFakeBench (Evaluation) |
> | --- | --- | --- | --- | --- | --- | --- |
> | Evaluation Data Size | 50,705 | 7,264 | 11,500 | 5,040 | 1,000 | 11,000 |
>
> > *Q3 It is unclear whether the misinformation types included are meaningful. For example, the Cross-modal Consistency Distortion example in Figure 2 — “An old man reading a book on a park bench” versus “An old man reading a newspaper on a park bench” — lacks practical relevance. Emphasis should be placed on edits involving named entities, such as celebrities, to ensure sufficient impact and deception. Similarly, the Visual Veracity Distortion example with “Veracious Text & AI-generated Image” lacks sufficient misleading potential; the reader’s first impression might be that the image is obviously edited, which fails to show it can lead viewers into perceiving falsified content as authentic.*
>
> Thanks. We have included **more visualized examples in Appendix Fig. 14, Fig. 15 and Fig. 16 of the updated paper**. Specifically, we address your concerns on the misinformation types in this work:
>
> - The examples in Fig. 2 are carefully selected to to **intuitively illustrate the key characteristics of misinformation** with reduced social harm. Furthermore, our dataset includes many examples with practical relevance and significant misleading potential, emphasizing its applicability to real-world misinformation scenarios. More visualized examples can be found in the in Fig. 14, Fig. 15 and Fig. 16 of the updated Appendix.
>
> - Inclusion of Celebrity-related edits. Our dataset encompasses misinformation scenarios involving celebrities, which is included in **celebrity-based query** for repurposed inconsistency, and **entity-edited artificial rumors**. In repurposed inconsistency, these cases create out-of-context mismatches by leveraging celebrities as query conditions. Additionally, our artificially generated rumors include entity substitutions, such as altered celebrity-related information.
>
> - Significance of the "Old Man Reading" Example. The example “Old man reading a book” versus “Old man reading a newspaper” in Figure 2 **exemplifies fine-grained inconsistencies**. It is specifically designed to **assess a model's ability to identify localized object-level differences**, such as distinguishing between a book and a newspaper. While seemingly simple, this example challenges models to detect subtle semantic conflicts, a capability essential for uncovering nuanced and sophisticated misinformation in professional fact-checking and real-world scenarios.

---

> ### Author Response · Authors · 2024-11-23
> **response to reviewer onkv (3)**
>
> > Q4 *For detecting outputs from generative models, a CNN often achieves better detection performance than an LVLM. Consider including performance results using foundation models paired with a fine-tuned network.*
>
> Thanks. We have included the experimental results of training foundation models paired with a fine-tuned network in **Appendix Table 9 (b) of the updated paper**. For convenience, we present the results in **Table r2**. Specifically, we selected existing powerful specialized detectors, FKA-Owl [r5] which integrates LVLMs with two learnable networks for fine-tuning. FKA-Owl is initially trained on the DGM4 [r1] and we fine-tune it on the MMFakeBench validation set incrementally with 10, 100, and 1000 examples. We report the results on both the DGM4 dataset and MMFakeBench test set. The results indicate that:
>
> (1) **Tuning-based model can be hard to generalize to unseen forgery data.** For off-the-shelf dedicated detectors without fine-tuning, their performance on the proposed benchmark dataset is notably poor. With the rapid development of generative models, new forgery techniques and synthesized data continue to emerge. Models trained on limited samples face significant challenges in generalizing to unseen types of forgery data, highlighting a critical issue in the field of fake detection [r10], [r11], [r5].
>
> (2) **Catastrophic forgetting issues are inevitable.** Fine-tuning on new data improves performance on MMFakeBench but simultaneously leads to a decline in performance on the original dataset. This degradation underscores the phenomenon of catastrophic forgetting, where previously acquired knowledge, such as that from the DGM4 dataset, is progressively lost.
>
> Table r2: Performance (F1 score ↑) on different models when under few-shot fine-tuning.
>
> |     | MMFakeBench | DGM4 |
> | --- | --- | --- |
> | FKA-Owl before fine-tuning | 44.6 | 78.5 |
> | FKA-Owl after fine-tuning using 10 examples | 46.9 | 78.3 |
> | FKA-Owl after fine-tuning using 100 examples | 61.1 | 66.4 |
> | FKA-Owl after fine-tuning using 1000 examples | 76.2 | 64.8 |
>
> > *Q5 Some details in Table 1 are unclear. For instance, why do MEIR and DGM4, which involve text editing, lack Textual Veracity Distortion? Edited texts are essentially a type of rumour.*
>
> Thanks. We would like to clarify that edited texts may sometimes contribute to rumors. **This depends on the intent behind the editing and whether the edited content has undergone verification**. Specifically, in the MEIR and DGM4 datasets, text editing is primarily designed to generate text-image inconsistencies rather than to create fact-conflicting content. Moreover, these edits are not subjected to fact-check verification by annotators. To provide greater clarity, we provide three examples from the DGM4 in Table r3 to illustrate situations where original and edited texts do not constitute a rumor. By judging the edited text in these examples, there is lack of sufficient evidence to debunk them.
>
> Additionally, recognizing that edited texts in MEIR and DGM4 could potentially produce rumor, **we have included circle-cross annotations in Table 1 of the updated paper** to clearly indicate datasets that unintentionally introduce unverified misinformation.
>
> Table r3: Examples of edited texts in DGM4 which do not bring misinformation.
>
> | Original Text | Edited Text |
> | --- | --- |
> | Many thousands of people **celebrate** the change on Monday. | Several thousand people **protested against** the change on Monday. |
> | Phone boxes in three villages have been **blocked** by their communitie. | Phone boxes in three villages have been **revamped** by their communities. |
> | Clinton of the Sanders campaign **praised** about me. | Clinton **so sick** of the Sanders campaign **lying** about me. |
>
> [r1] Shao R, et al. Detecting and Grounding Multi-Modal Media Manipulation. CVPR, 2023.
>
> [r2] Long L, et al. On LLMs-Driven Synthetic Data Generation, Curation, and Evaluation: A Survey. 2024.
>
> [r3] Zeng F, et al. Multimodal Misinformation Detection by Learning from Synthetic Data with Multimodal LLMs. EMNLP Findings, 2024.
>
> [r4] Qi P, et al. SNIFFER: Multimodal Large Language Model for Explainable Out-of-Context Misinformation Detection. CVPR, 2024.
>
> [r5] Liu X, et al. FKA-Owl: Advancing Multimodal Fake News Detection through Knowledge-Augmented LVLMs. ACM MM, 2024.
>
> [r6] Luo G, et al. NewsCLIPpings: Automatic Generation of Out-of-Context Multimodal Media. EMNLP, 2021.
>
> [r7] Yue X, et al. MMMU: A Massive Multi-discipline Multimodal Understanding and Reasoning Benchmark for Expert AGI. CVPR, 2024.
>
> [r8] Liu X, et al. MM-SafetyBench: A Benchmark for Safety Evaluation of Multimodal Large Language Models. ECCV, 2024.
>
> [r9] Liu Y, et al. OCRBench: On the Hidden Mystery of OCR in Large Multimodal Models. 2024.
>
> [r10] Akhtar M, et al. MDFEND: Multi-domain Fake News Detection. CIKM, 2021.
>
> [r11] Ojha U, et al. Towards Universal Fake Image Detectors That Generalize Across Generative Models. CVPR, 2023.

---

> ### Author Response · Authors · 2024-11-25
> **Looking forward to the response from Reviewer onkv**
>
> Dear Reviewer onkv,
>
> We have tried our best to address all the concerns and provided as much evidence as possible. May we know if our rebuttals answer all your questions? We truly appreciate it.
>
> Best regards,
>
> Author #6524

---

### Official Review · Reviewer_yehn · 2024-11-03

**Soundness:** 3
**Presentation:** 3
**Contribution:** 3
**Rating:** 6
**Confidence:** 4

**Summary:**

This work proposes the first benchmark in mixed-source multimodal misinformation detection (MMD). The authors provide comprehensive evaluations and analysis under the proposed evaluation settings, providing a new baseline for future research.

**Strengths:**

1. It is important to create such a benchmark and baseline for the detection of multi-source fakes.

2. The proposed three types based on the sources of falsified content are new to the field and technical sound.

3. The designed MMD-Agent is well-designed and novel.

**Weaknesses:**

1. Lack of comparison with related detection methods: This work includes two detectors within the visual veracity distortion (VDD) field, both relying solely on visual modules for detection. Why not include comparisons with recent methods that integrate both visual and language modules, which are more directly relevant to this research?

2. Insufficient comparison with deepfake detection methods: If "misinformation" encompasses any fake content, it would be beneficial for the authors to compare their method with established deepfake detectors. Given the high profile and potential danger of deepfakes, adding such comparisons would strengthen the paper’s impact and comprehensiveness.

3. Lack of intuitive visualizations for the proposed method: It is recommended that the authors use visualization tools like GradCAM to provide clearer, more intuitive insights into their detection results. Additionally, the current t-SNE visualizations could be improved to more clearly convey their underlying meaning.

**Questions:**

1. What is the precise scope of this work? For example, does deepfake detection fall within its scope? If so, how does the work address multi-modal detection challenges, such as face reenactment or face swapping?

2. How do Vision-Language Models (VLMs), which incorporate both language and vision modules, perform within the proposed benchmark?

More questions can be seen in the "Weakness" part.

**Details Of Ethics Concerns:**

It appears that the authors directly use and collect real data from the original datasets. However, they do not specify whether they have obtained permission (or copyright) to distribute this real data or mention any measures in place to ensure its legal distribution.

---

> ### Author Response · Authors · 2024-11-23
> **response to reviewer yehn (1)**
>
> Thanks for the thorough review and constructive feedback on our paper. We appreciate the recognition of the importance of the MMFakeBench benchmark, particularly serving as a baseline for the detection of multiple-source fakes. Below, we address the specific concerns raised in the review:
>
> > *Q1 Lack of comparison with related detection methods: Why not include comparisons with recent methods that integrate both visual and language modules.*
>
> Thanks. We would like to clarify that the comparisons with methods integrating both visual and language models have already been included in the **Table 3 of the main paper**. For convenience, we present the required results **in Table r1**. These multimodal method evaluation specifically examines the performance of FakeNewsGPT4 [r1] and HAMMER [r2]. The results show that the LLaVA-1.6-34B model performs better than single-source detectors for both binary and multiclass classification by a large margin. This performance gap highlights a critical challenge in specialized detectors: tuning-based models often struggle to generalize to unseen forgery methods and data, which serve as a critical issue in fake detection [r3], [r4], [r1]. These findings highlight that LVLMs can serve as potential general detectors in achieving robust detection of multimodal misinformation.
>
> Table r1: Performance (F1 score ↑) comparison of selected models on the proposed dataset.
>
> |     | F1 (Binary) | F1 (Multiclass) |
> | --- | --- | --- |
> | FakeNewsGPT4 | 41.7 | /   |
> | HAMMER | 43.0 | /   |
> | LLaVA-1.6-34B | 67.2 | 49.9 |
>
> > *Q2 Insufficient comparison with deepfake detection methods*
>
> Thanks. As suggested, we conduct experiments using three representative deepfake detection methods, including **Core** [r5], **RECCE** [r6], and **UCF** [r7], on our proposed dataset. The results are summarized in **Table r2. These methods, while performing well on existing deepfake datasets, showed significantly reduced generalization capabilities when applied to our dataset. We attribute this observation to the following factors:
>
> - **Mixed-source Nature of Our Dataset**. Unlike traditional deepfake datasets, which primarily focus on image-based forgeries, our dataset introduces forgeries across multiple modalities, including textual veracity, visual veracity, and cross-modal consistency. For instance, a given sample may involve false textual claims paired with real images or repurposed inconsistencies between the real text and real image. This mixed-source complexity poses additional challenges that single-modal deepfake detection methods are not designed to address.
>
> - **Challenging Generalization.** Deepfake detection methods often perform well on datasets and forgery techniques they have been trained on. However, their generalization capability remains challenging to unseen, AI-generated content [r8], [r9], particularly when datasets are diverse and forgery methods are novel or underexplored. Our dataset includes samples generated using a wide range of advanced generative techniques, which represent more complex and diverse scenarios than conventional deepfake datasets, further exposing the limitations of existing methods.
>
>
> Table r2: Performance (F1 score ↑) of deepfake detection methods on the proposed benchmark dataset.
>
> |     | F1 (Binary) | F1 (Multiclass) |
> | --- | --- | --- |
> | Core | 44.5 | /   |
> | RECCE | 46.9 | /   |
> | UCF | 44.9 | /   |
> | LLaVA-1.6-34B | 67.2 | 49.9 |
>
> > *Q3 Lack of intuitive visualizations for the proposed method. Additionally, the current t-SNE visualizations could be improved to more clearly convey their underlying meaning.
>
> Thanks. We have included LVLM inpretation results of the proposed method in **Appendix Fig. 9 of the updated paper** and improved t-SNE visualizations in **Appendix Fig. 10 of the updated paper**.
>
> - **Interpretation Results.** We have incorporated the recent LVLM-interpret [r10] as a visualization tool to provide a more transparent understanding of the model's decision-making process. This method adapts the calculation of relevancy maps to LVLM, thus providing a detailed representation of the regions of the image most relevant to each generated token. Specifically, as shown in **Appendix Fig. 10 (e) of the updated paper** for cross-modal consistency distortion, the relevancy map focuses on the clapping action of the characters in the image. Clapping typically signifies recognition or approval of an event, conveying semantics that differ from the meaning of the associated text. This observation aligns with the model's detection output, providing intuitive explanations for the identified inconsistency.
>
> - **Improved t-SNE Visualizations.** We include color-coded annotations that correspond to specific forgery sources, making it easier to interpret the distribution of samples. The revised visualizations better illustrate that each distinct source is distinctly dispersed across the feature space, revealing the diversity of our proposed dataset.

---

> ### Author Response · Authors · 2024-11-23
> **response to reviewer yehn (2)**
>
> > *Q4 What is the precise scope of this work? For example, does deepfake detection fall within its scope? If so, how does the work address multi-modal detection challenges, such as face reenactment or face swapping?
>
> Thanks. We address your concerns concerning the precise scope of this work as follows:
>
> - **The scope of this work**. Our dataset is specifically designed as a **content-centric** multimodal misinformation benchmark. Its primary objective is to evaluate the veracity of visual and textual content, as well as the consistency between image and text modalities.
>
> - **The Characteristics of the Deepfake methods**. Deepfake detection methods primarily focus on identifying **content-agnostic artifact traces**, which is why this work **does not** **assign a specific forgery type exclusively for deepfakes**. In fact, our dataset includes examples of abnormal facial manipulations in PS-edited images and AI-generated scenarios featuring celebrities in fictional contexts. More visualized examples of visual veracity distortions can be found in **Appendix Fig. 15 of the updated paper**.
>
> - **Unique** mixed-source multimodal misinformation detection benchmark. There are a wealth of existing datasets [r11], [r12] and benchmarks [13] dedicated to deepfake scenarios. In contrast, our proposed multimodal misinformation benchmark addresses a significant gap in the field by focusing on mixed-source misinformation. If incorporating deepfake-related works is necessary, existing datasets can be utilized to supplement such requirements.
>
>
> > *Q5 How do Vision-Language Models (VLMs), which incorporate both language and vision modules, perform within the proposed benchmark?
>
> Thanks. We would like to clarify that VLMs perform within the proposed benchmark have already been included in the **Table 2 and** **Appendix Table 9 of the paper**. For convenience, we present the results in **Table r3**. The evaluated LVLMs in our paper, such as LLaVA, inherently integrate both visual and language modules, and thus can be considered a subset of VLM models. To ensure a more comprehensive evaluation, we also conducted experiments on VLM models that are not based on large-scale foundational models. Specifically, we perform experiments on the proposed benchmark on the HAMMER [r2] model, which utilizes a traditional dual-tower VLM architecture, ALBEF [r14]. HAMMER is initially trained on the DGM4 [r2] and we fine-tune it on the MMFakeBench validation set incrementally with 10, 100, and 1000 examples. We report the results on both the DGM4 dataset and the MMFakeBench test set. The results indicate that:
>
> (1) **Tuning-based models are hard to generalize to unseen forgery data.** For off-the-shelf dedicated detectors without fine-tuning, their performance on the proposed benchmark dataset is notably poor. With the rapid development of generative models, new forgery techniques and synthesized data continue to emerge. Models trained on limited samples face significant challenges in generalizing to unseen types of forgery data, highlighting a critical issue in the field of fake detection [r3], [r4], [r1].
>
> (2) **Catastrophic forgetting issues are inevitable.** Fine-tuning on new data improves performance on MMFakeBench but simultaneously leads to a decline in performance on the original dataset. This degradation underscores the phenomenon of catastrophic forgetting, where previously acquired knowledge, such as that from the DGM4 dataset, is progressively lost.
>
> Table r3: Performance (F1 score ↑) on different models when under few-shot fine-tuning.
>
> |     | DGM4 | MMFakeBench |
> | --- | --- | --- |
> | HAMMER before tuning | 83.2 | 44.1 |
> | HAMMER after tuning on 10 | 78.7 | 46.8 |
> | HAMMER after tuning on 100 | 70.1 | 60.8 |
> | HAMMER after tuning on 1000 | 63.2 | 72.4 |

---

> ### Author Response · Authors · 2024-11-23
> **response to reviewer yehn (3)**
>
> > Q6 It appears that the authors directly use and collect real data from the original datasets. However, they do not specify whether they have obtained permission (or copyright) to distribute this real data or mention any measures in place to ensure its legal distribution.
>
> Thanks. The real dataset is constructed by sampling from Visual News [r15], MS-COCO [r16], and Fakeddit [r17]. We would like to clarify that we have proactively contacted the respective authors of the datasets to obtain explicit permission for their integration and potential distribution within our work. These communications are conducted via email to ensure proper documentation and full compliance with copyright and licensing requirements.
>
> [r1] Liu X, et al. FKA-Owl: Advancing Multimodal Fake News Detection through Knowledge-Augmented LVLMs. ACM MM, 2024.
>
> [r2] Shao R, et al. Detecting and Grounding Multi-Modal Media Manipulation. CVPR, 2023.
>
> [r3] Akhtar M, et al. MDFEND: Multi-domain Fake News Detection. CIKM, 2021.
>
> [r4] Ojha U, et al. Towards Universal Fake Image Detectors That Generalize Across Generative Models. CVPR, 2023.
>
> [r5] Ni Y, et al. CORE: COnsistent REpresentation Learning for Face Forgery Detection. CVPRW, 2022.
>
> [r6] Cao J, et al. End-to-End Reconstruction-Classification Learning for Face Forgery Detection. CVPR, 2022.
>
> [r7] Yan Z, et al. UCF: Uncovering Common Features for Generalizable Deepfake Detection. ICCV, 2023.
>
> [r8] Yao K, et al. Towards Understanding the Generalization of Deepfake Detectors from a Game-Theoretical View. ICCV, 2023.
>
> [r9] Lin L, et al. Preserving Fairness Generalization in Deepfake Detection. CVPR, 2024.
>
> [r10] Gabriela Ben Melech Stan, et al. LVLM-Interpret: An Interpretability Tool for Large Vision-Language Models. 2024.
>
> [r11] Rossler A, et al. Faceforensics++: Learning to detect manipulated facial images. CVPR, 2019.
>
> [r12] Li Y, et al. Celeb-df: A new dataset for deepfake forensics. CVPR, 2020.
>
> [r13] Yan Z, et al. DeepfakeBench: A Comprehensive Benchmark of Deepfake Detection. NeurIPS, 2023.
>
> [r14] Li J, et al. Align before Fuse: Vision and Language Representation Learning with Momentum Distillation. NeurIPS, 2021.
>
> [r15] Liu F, et al. Visual News: Benchmark and Challenges in News Image Captioning. EMNLP, 2021.
>
> [r16] Lin T, et al. Microsoft COCO: Common Objects in Context, 2015.
>
> [r17] Nakamura K, et al. r/Fakeddit: A New Multimodal Benchmark Dataset for Fine-grained Fake News Detection. LREC, 2020.

---

> ### Author Response · Authors · 2024-11-25
> **Looking forward to the response from Reviewer yehn**
>
> Dear Reviewer yehn,
>
> We have tried our best to address all the concerns and provided as much evidence as possible. May we know if our rebuttals answer all your questions? We truly appreciate it.
>
> Best regards,
>
> Author #6524

---

### Official Review · Reviewer_jzRM · 2024-11-03

**Soundness:** 3
**Presentation:** 4
**Contribution:** 3
**Rating:** 8
**Confidence:** 4

**Summary:**

This paper looks at creating a multimodal misinformation dataset called MMFakeBench that can be used to evaluate detection methods and Large-Vision Models (LVLMs) under a zero-shot setting. For the dataset a validation set is created for hyperparameter selection and also used to conduct a human evaluation of the proposed MMFakeBench dataset. Utilizing the MMFakeBench dataset, 15 LVLMs are evaluated on the test set of the dataset and highlights the shortcomings of these LVLMs when compared to the human evaluation results.

**Strengths:**

* Utilizing a human evaluation helped strengthen the claim that currently LVLMs models have room for improvement when it comes to tackling the tasks proposed with the MMFakeBench dataset.
* Table 2 was a well thought out table that coverages a wide range of LVLMs
* The proposed dataset is unique as it proposes different tasks related to mixed-source multimodal misinformation detection, as it shows in Table 1 the difference between its dataset and previous multimodal misinformation datasets.

**Weaknesses:**

* Currently the dataset size seems somewhat low when comparing it to other datasets mentioned in the paper. However I do recognize that this is specifically meant to be an evaluation dataset.

**Questions:**

* Can the authors possibly give further examples of evaluation type datasets that look at evaluation in a zero-shot setting for misinformation detection.

---

> ### Author Response · Authors · 2024-11-23
> **response to reviewer jzRM (1)**
>
> Thank you for your thoughtful and constructive feedback. We are encouraged that you find our proposed dataset is unique compared to existing datasets and provide thorough experimental results for evaluating LVLMs. Below are our responses to your concerns.
>
> > *Q1 Currently the dataset size seems somewhat low when comparing it to other datasets mentioned in the paper. However I do recognize that this is specifically meant to be an evaluation dataset.*
>
> Thanks. We sincerely appreciate your acknowledgment that our dataset has been specifically designed as an evaluation benchmark. As an evaluation dataset, MMFakeBench is **comparable in scale to other LVLM-based evaluation datasets** (such as a representative dataset MMMU [r1] with 11,500 samples) and is well-suited for assessing model performance. Although its size is not exceptionally large, we have meticulously designed the dataset to **ensure comprehensive diversity**, encompassing a wide range of multimodal misinformation types. This diversity is achieved by identifying three critical sources of multimodal misinformation and incorporating twelve distinct forgery types, each driven by varying intents, themes, and tools. For each forgery type, we carefully examine and incorporate different generative techniques to simulate real-world misinformation scenarios. The result is a comprehensive and challenging dataset that serves as a valuable resource for advancing research in the critical field of multimodal misinformation detection.

---

> ### Author Response · Authors · 2024-11-23
> **response to reviewer jzRM (2)**
>
> > *Q2 Can the authors possibly give further examples of evaluation type datasets that look at evaluation in a zero-shot setting for misinformation detection.*
>
> Thanks. We have conducted an experiment to compare the performance of selected models on the proposed benchmark against existing benchmarks (datasets) in **Appendix Section A.1.4 of the paper**. For convenience, we have presented results in **Table r1**. Additonally, we have provide more evaluation examples to conduct a deep analysis in **Appendix Section A.1.7 of the updated paper**.
>
> - **Evaluation results on existing datasets.** Two widely-used multimodal misinformation datasets, **NewsClippings** [r2] and **Fakeddit** [r3] have been utilized to evaluate LVLMs [r4], [r5]. We employ four models: two powerful open-source LVLM models, LLaVA-1.6-34B and BLIP2-FLANT5-XXL with two open-source multimodal specialized detectors, HAMMER [r6] and FKA-Owl [r7]. All these models are evaluated on two multimodal misinformation datasets, NewsClippings and Fakeddit. The results indicate that our MMFakeBench poses a greater challenge in binary classification compared to the other two datasets. More importantly, the primary challenge of our benchmark lies in **its capacity to perform fine-grained assessments of misinformation sources in real-world scenarios where multiple forgery types coexist**. This mirrors the complexity of real-world environments, where misinformation stems from diverse, overlapping sources. This capability is essential for addressing real-world challenges and underscores the importance of MMFakeBench in advancing multimodal misinformation detection.
> - **More case studies**. We have presented more case studies that span four models of different scales and on three distinct forgery sources. The key findings are summarized as follows: (1) Existing models exhibit **reliance on external knowledge** and are challenging to infer the factual correctness of a statement without access to a broader context. (2) Existing LVLM models GPT-4V face **challenges in discerning abnormal physical characteristics**. (3) When addressing subtle inconsistencies, existing models **may be distracted by the dominant presence of consistent content**, thereby failing to discern subtle mismatches critical for accurate misinformation detection.
>
> Table r1: Performance (F1 score ↑) comparison of selected models on the proposed benchmark and existing benchmarks (datasets).
>
> |     | NewsCLIPpings (Binary) | NewsCLIPpings (Multiclass) | Fakeddit (Binary) | Fakeddit (Multiclass) | MMFakeBench (Binary) | MMFakeBench (Multiclass) |
> | --- | --- | --- | --- | --- | --- | --- |
> | BLIP2-FLANT5-XXL | 33.6 | -   | 40.5 | -   | 31.6 | 16.7 |
> | LLaVA-1.6-34B | 65.6 | -   | 71.4 | -   | 62.9 | 25.7 |
> | FKA-Owl | 52.0 | -   | 55.0 | -   | 41.7 | -   |
> | HAMMER | 55.0 | -   | 52.6 | -   | 43  | -   |
>
> [r1] Yue X, et al. MMMU: A Massive Multi-discipline Multimodal Understanding and Reasoning Benchmark for Expert AGI. CVPR, 2024.
>
> [r2] Luo G, et al. NewsCLIPpings: Automatic Generation of Out-of-Context Multimodal Media. EMNLP, 2021.
>
> [r3] Nakamura K, et al. r/Fakeddit: A New Multimodal Benchmark Dataset for Fine-grained Fake News Detection. LREC, 2020.
>
> [r4] Qi P, et al. SNIFFER: Multimodal Large Language Model for Explainable Out-of-Context Misinformation Detection. CVPR, 2024.
>
> [r5] Xuan K, et al. LEMMA: Towards LVLM-Enhanced Multimodal Misinformation Detection with External Knowledge Augmentation. 2024.
>
> [r6] Shao R, et al. Detecting and Grounding Multi-Modal Media Manipulation. CVPR, 2023.
>
> [r7] Liu X, et al. FKA-Owl: Advancing Multimodal Fake News Detection through Knowledge Augmented LVLMs. ACM MM, 2024

---

> > ### Comment · Reviewer_jzRM · 2024-11-26
> >
> > Thank you for your detailed response! I thought this work was quite interesting to read and is an important field of study considering the wide utilization of LVLM's for varying tasks and it is interesting to see how they work in this setting. Most of my concerns are met I recommend acceptance . Good luck in the review process!

---

### Author Response · Authors · 2024-11-23
**general response to reviewers**

Dear reviewers,

We would like to express our sincere gratitude for your invaluable suggestions and patience regarding our work. We appreciate the recognition of key contributions in this work by (reviewers jzRM, yehn, x4j6, and RALP): addressing the real-world critical yet underexplored issue of the con-existence of multiple forgery sources, and proposes the first benchmark explicitly designed for evaluating mixed-source multimodal misinformation.

We have been dedicated to enhancing our paper based on your feedback. We have improved our paper in the following aspects:

- Updated Experiments:

  - Conducting ablation experiments on external knowledge sources.

  - Fine-tuning vision-language detectors on the proposed dataset.

- Revisions Based on Feedback: We have made thoughtful revisions in line with your suggestions, corrections in citations, references, experimental analysis, etc.

- Enhanced Result Presentation:

  - Providing more error analysis.

  - Providing interpretable visualization of the model's decision-making process.

  - Improved TSNE visualization.

  - Providing more visualization examples within the dataset.


**All revision text is marked with blue.** We believe these revisions significantly strengthen our paper and hope they adequately address your concerns and suggestions. We look forward to your further feedback and thank you once again for your valuable contributions to our work.

Best regards,

Authors of paper submission 6524

---

### Author Response · Authors · 2024-12-02
**Thanks for the time and efforts invested by all reviewers and AC**

Dear AC and Reviewers,

Firstly, we wish to express our profound gratitude for your time, effort, and insightful feedback on our manuscript. We deeply appreciate the recognition our work has received: Reviewer jzRM and Reviewer RALP emphasize the novelty of our proposed benchmark, Reviewer yehn acknowledges its significant contribution, Reviewer onkv highlights the comprehensive types covered by our benchmark and Reviewer x4j6 praises the state-of-the-art performance by the proposed method.

Throughout the review process, several concerns are raised, primarily related to technical details, enhanced result presentation, and the necessity for additional ablations and comparisons. We have addressed most of these concerns during the rebuttal stages.

The reviewer onkv raises **concerns regarding the novelty of our work**, to which we have responded with detailed clarifications, but unfortunately, we have not received further feedback. To address this, we would like to highlight two points here. Firstly, our work addresses the **real-world critical yet underexplored issue** of the con-existence of multiple forgery sources. These mixed-source scenarios necessitate detection methods capable of generalizing **beyond single-source constraints**, which **existing datasets and detectors fail to support**. Secondly, **we emphasize the importance and comprehensiveness of our dataset**. Our MMFakeBench is the **first** benchmark explicitly designed for **evaluating mixed-source multimodal misinformation**, encompassing three primary forgery categories and twelve subcategories of forgery types.

We believe that this mixed-source multimodal misinformation benchmark proposed in our paper are of significant value to this field, marking a significant step toward addressing misinformation in practical applications.

We sincerely appreciate the time and efforts invested by all reviewers and AC in evaluating our work.

Best regards,

Authors of paper submission 6524

---

### Meta-Review · Area_Chair_8hJP · 2024-12-19

**Metareview:**

This work introduces a mixed-source multimodal misinformation detection dataset (MMD). The authors provide comprehensive evaluations on the proposed evaluation settings, and introduce a new baseline for this dataset. Most of the reviewers have agreed that the authors have successfully addressed their concerns and raised their initial ratings. Overall, the comments of the reviewers are positive, and based on the positive comments and the increased ratings after rebuttal, AC recommends to accept this work.

**Additional Comments On Reviewer Discussion:**

Most of the reviewers agreed their concerns have been fully addressed by the authors. There is no strong argument to reject this paper. Although reviewer x4j6 still kept the initial rating as the rebuttal only address the concerns partially, AC reads the rebuttal and thinks the rebuttal is convincing.

---

### Decision · Program_Chairs · 2025-01-22

Accept (Poster)